# DrVD-Bench: Do Vision-Language Models Reason Like Human Doctors in Medical Image Diagnosis?

**Tianhong Zhou**[1*]   **Yin Xu**[1*]   **Yingtao Zhu**[1*]   **Chuxi Xiao**[1]
**Haiyang Bian**[2†]   **Lei Wei**[1†]   **Xuegong Zhang**[1]
[1]Tsinghua University
[2]Alibaba Cloud Computing

## Abstract

Vision–language models (VLMs) exhibit strong zero-shot generalization on natural images and show early promise in interpretable medical image analysis. However, existing benchmarks do not systematically evaluate whether these models truly reason like human clinicians or merely imitate superficial patterns. To address this gap, we propose DrVD-Bench, the first multimodal benchmark for clinical visual reasoning. DrVD-Bench consists of three modules: *Visual Evidence Comprehension*, *Reasoning Trajectory Assessment*, and *Report Generation Evaluation*, comprising a total of 7,789 image–question pairs. Our benchmark covers 20 task types, 17 diagnostic categories, and five imaging modalities—CT, MRI, ultrasound, radiography, and pathology. DrVD-Bench is explicitly structured to reflect the clinical reasoning workflow from modality recognition to lesion identification and diagnosis. We benchmark 19 VLMs, including general-purpose and medical-specific, open-source and proprietary models, and observe that performance drops sharply as reasoning complexity increases. While some models begin to exhibit traces of human-like reasoning, they often still rely on shortcut correlations rather than grounded visual understanding. DrVD-Bench offers a rigorous and structured evaluation framework to guide the development of clinically trustworthy VLMs.
**Dataset:** Kaggle, Hugging Face     **Code:** GitHub

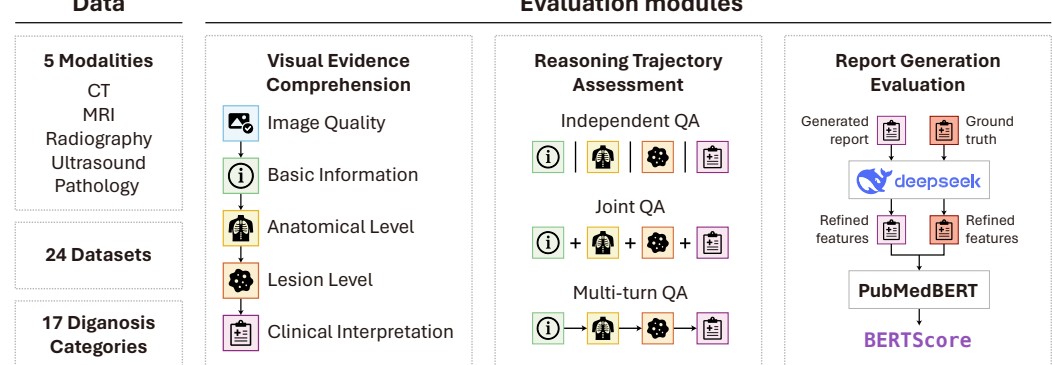

Figure 1: Overview of the DrVD-Bench

---

[*]Equal contribution.
[†]Corresponding authors: `bianhaiyang.bhy@alibaba-inc.com`, `weilei92@tsinghua.edu.cn`

39th Conference on Neural Information Processing Systems (NeurIPS 2025) Track on Datasets and Benchmarks.

# 1 Introduction

The lack of interpretability in AI-driven diagnostic systems has long been a critical barrier to their adoption in clinical practice [87, 36]. Vision-Language Models (VLMs) [29, 18, 14, 27, 74, 35, 90] offer a promising direction toward interpretable medical AI, as they can generate outputs that mimic the stepwise clinical reasoning process of physicians [72, 13, 66]. In clinical settings, physicians arrive at diagnosis by progressively integrating information across multiple levels—ranging from basic image features to anatomical structures and lesion characteristics—before synthesizing this evidence step-by-step into a clinical diagnostic report. Despite the increasing adoption of VLMs in analyzing medical images, there exists no benchmark that systematically evaluates whether these models follow a similar reasoning trajectory or merely rely on memorized patterns and spurious shortcuts.

To address this gap, we introduce DrVD-Bench (Doctor-like Visual Diagnosis Benchmark), the first benchmark explicitly designed to evaluate the understanding and reasoning capabilities of VLMs in medical image analysis. DrVD-Bench comprises three complementary modules: (i) *Visual Evidence Comprehension*, containing 4,480 high-quality image–question pairs structured to mirror the each step of clinical reasoning, assessing how well models identify essential visual cues and generate diagnoses; (ii) *Reasoning Trajectory Assessment*, consisting of 487 images with 3,321 question–answer turns that emulate the progressive reasoning process of clinicians and evaluate whether models reason step by step or rely on shortcuts; and (iii) *Report Generation Evaluation*, comprising 475 questions aimed at assessing holistic understanding through free-form clinical report generation.

Modules 1 and 2 are explicitly designed to reflect the clinical reasoning workflow—from modality recognition to anatomical localization, lesion characterization, and final diagnosis. This workflow is defined as follows:

- **Level 0: Image Quality Assessment** — Detecting artifacts or noise; typically handled during acquisition.

- **Level 1: Basic Information Extraction** — Identifying modality, body region, view, etc.; often available as metadata in clinical settings.

- **Level 2: Anatomy-Level Recognition** — Recognizing and localizing anatomical structures (e.g., organs, tissues).

- **Level 3: Lesion-Level Identification** — Detecting, localizing, and describing abnormalities such as lesions or fractures.

- **Level 4: Clinical Interpretation** — Integrating visual findings to produce a diagnosis or generate a structured report.

By organizing tasks into stepwise levels, DrVD-Bench enables fine-grained analysis of VLM performance—not only in terms of final diagnostic predictions, but also in their capacity to extract intermediate visual evidence and engage in coherent reasoning. This structure allows us to examine whether models genuinely analyze medical images by identifying key visual cues and reasoning in a step-by-step manner, akin to human clinicians. In the *Visual Evidence Comprehension* module, we introduce specially designed *organ erasure* and *lesion erasure* tasks, which compel models to rely on visible image content rather than memorized associations, thereby mitigating potential information leakage from pretraining. The *Reasoning Trajectory Assessment* module includes three distinct QA formats—*Independent QA*, *Joint QA*, and *Multi-turn QA*, —with the latter specifically enforcing a stepwise reasoning pattern that mirrors clinical workflows. Except for this module, all other stages adopt an independent QA format, where each question is presented in isolation without shared context. Finally, unlike previous benchmarks that limit report generation to specific modalities [41, 75, 93] such as radiographs or CT scans, our *Report Generation Evaluation* module introduces a setting to assess models' holistic understanding across diverse imaging modalities.

We evaluated 19 publicly available VLMs, including both general purpose and medical-specific models (see Table 3). Our key findings are summarized as follows:

- **Reasoning performance declines with task complexity:** VLMs perform well on low-level visual tasks such as modality or view recognition, but their accuracy drops substantially on higher-level tasks involving anatomical understanding, lesion localization, and diagnosis.

- **Correct answers without supporting evidence:** Many models achieve higher diagnostic accuracy than their performance on lesion-level tasks, suggesting they can produce correct diagnoses without fully understanding or localizing the supporting visual evidence.

- **Limited capacity for stepwise clinical reasoning:** Models perform best when provided with all questions at once (Joint QA), but struggle in Multi-turn QA, suggesting difficulty with maintaining dialogue state and reasoning trajectories.

- **Hallucinations in report generation:** In free-form generation tasks, models often produce plausible but unsupported statements, revealing challenges in grounding clinical language in image evidence.

- **Smaller, specialized models can compete:** While larger and newer models generally perform better, domain-optimized models demonstrate strong performance relative to their scale, highlighting the value of medical-specific alignment.

## 2 Related Works

### 2.1 Vision-Language Models

Modern vision-language models (VLMs) build upon the reasoning capabilities and world knowledge of large language models (LLMs) by aligning visual inputs with the textual domain [90]. For instance, LLaVA [45] introduces a multi-layer perceptron (MLP)[59] to bridge a vision encoder (e.g., CLIP [62]) with a language model backbone, enabling the system to perform tasks such as interpreting scientific figures [82] and understanding cartoons [47]. VLMs have also been adapted for the medical domain [42, 89, 39]. For example, LLaVA-Med [39], derived from LLaVA, is fine-tuned on medical data to adapt the model for healthcare-specific tasks. Although VLMs have shown impressive performance on general visual reasoning tasks [16, 88, 32, 51], it remains uncertain whether they truly comprehend medical images or merely rely on prior knowledge and pattern matching [17, 77, 60, 37].

### 2.2 Medical VLM Benchmarks

The application of VLMs in medicine demands benchmarks with broad coverage and fine-grained, clinically relevant evaluation. However, most existing benchmarks remain narrow in scope (see Table 1), typically limited to single modalities or task types. For example, PathMMU [73], VQA-RAD [38], and PMC-VQA [92] focus primarily on visual question answering (VQA) within specific imaging domains, limiting generalizability and diagnostic depth.

Recent benchmarks introduce hierarchical structures [58, 85, 31], but these do not align with the stepwise nature of clinical reasoning. GMAI-MMBench [85], for instance, organizes questions by perceptual complexity rather than reasoning stages. OmniMedVQA [31] covers diverse tasks like modality recognition and diagnosis but lacks a clinically grounded task progression, grouping questions only by type. As a result, current benchmarks fall short in evaluating whether VLMs reason like clinicians. A critical gap remains: we lack a benchmark that not only measures answer correctness but also reveals how and why models succeed or fail—essential for assessing clinical reasoning ability.

Table 1: Comparison of Medical VLM Benchmarks

| Benchmark | Imaging modalities | Task hierarchy | Clinical reasoning | Task types |
|---|---|---|---|---|
| VQA-RAD[38] | Radiography, CT | ✗ | ✗ | VQA |
| SLAKE[44] | Radiography, CT, MRI | ✗ | ✗ | VQA |
| PMC-VQA [92] | CT, MRI, and others | ✗ | ✗ | VQA |
| Rad-ReStruct [58] | Radiography | ✓ | ✗ | VQA |
| GMAI-MMBench[85] | CT, MRI, Radiography, Ultrasound, Pathology | ✓ | ✗ | VQA |
| PathMMU[73] | Pathology | ✗ | ✗ | VQA |
| OmniMedVQA[31] | 12 modalities | ✓ | ✗ | VQA |
| CARES[79] | 16 modalities | ✗ | ✗ | VQA |
| MultiMedEval[68] | ≥ 11 modalities | ✗ | ✗ | VQA, open QA, and others |
| **DrVD-Bench** | 5 modalities | ✓ (stepwise) | ✓ | VQA, report generation |

# 3 Design of DrVD-Bench

## 3.1 Overview

We propose DrVD-Bench, a multi-scale benchmark for systematically evaluating vision–language models (VLMs) in the medical domain. Inspired by the diagnostic workflow of clinicians, DrVD-Bench defines a three-module framework that evaluates VLMs from three aspects: (i) Reliability in visual evidence extraction; (ii) Ability in stepwise clinical reasoning; and (iii) Comprehensive understanding of medical image-revealed by the ability in report generation. Representative examples are shown in Figure 2.

DrVD-Bench comprises three modules: (1) **Visual Evidence Comprehension**, containing 4,480 expert-curated image–question pairs across 16 tasks, structured by the depth of clinical reasoning, from the superficial modality recognition to the deep lesion-level identification and diagnosis. To reduce reliance on shortcuts, we introduce **organ** and **lesion erasure** tasks that force models to reason from visible evidence. (2) **Reasoning Trajectory Assessment**, with 3,321 QA turns on 487 images, evaluates whether models reason step by step using three prompting formats: **Joint QA**, **Independent QA**, and **Multi-turn QA**. (3) **Report Generation Evaluation**, spanning all the five modalities, it contains 475 questions aimed at assessing holistic understanding through free-form clinical report generation. See Appendix A.2 for the detailed composition of our benchmark.

Together, these components enable fine-grained analysis of VLMs' clinical visual understanding and diagnostic reasoning.

## 3.2 Dataset Collection and Task Construction

### 3.2.1 Data Collection

This study systematically aggregates multi-modal medical images (CT, MRI, ultrasound, X-ray, and pathology) from 24 publicly available datasets and online repositories (See Table 8 in Appendix A ). These sources encompass a wide range of imaging scales, including panoramic, organ-level, and histopathological views. Only images with a resolution of at least $256 \times 256$ are retained to ensure sufficient visual clarity for structural and anatomical interpretation;

### 3.2.2 Task Overview

To enable fine-grained evaluation of models' intermediate reasoning in medical image analysis, DrVD-Bench organizes all QA tasks (except report generation) into a five-level hierarchy that mirrors the cognitive stages of clinical diagnosis. Each level targets a distinct reasoning step:

- **Level 0 (Image Quality):** noise and artifact detection.
- **Level 1 (Basic Information):** modality, view, body part, magnification, stain, and imaging technique recognition.
- **Level 2 (Anatomy Level):** organ/tissue identification, localization, and organ-erasure detection.
- **Level 3 (Lesion Level):** lesion recognition, lesion-erasure detection, and morphological description.
- **Level 4 (Clinical Interpretation):** diagnostic classification.

The *Visual Evidence Comprehension* module includes tasks from all levels, with each image paired with a single QA focused on one aspect of visual understanding. In contrast, the *Reasoning Trajectory Assessment* module evaluates multi-level reasoning per image by combining one task from each level (e.g., modality, body part, organ, lesion, diagnosis). It offers three prompting formats: **Joint QA** presents all questions at once; **Independent QA** asks them sequentially; and **Multi-turn QA** incorporates the model's prior response into subsequent prompts, simulating stepwise reasoning.

### 3.2.3 Dataset and QA Pair Construction

To support diverse evaluation objectives, we employ task-specific dataset construction strategies. For noise robustness, we simulate modality-specific clinical noise at three PSNR levels—15, 25, and 35

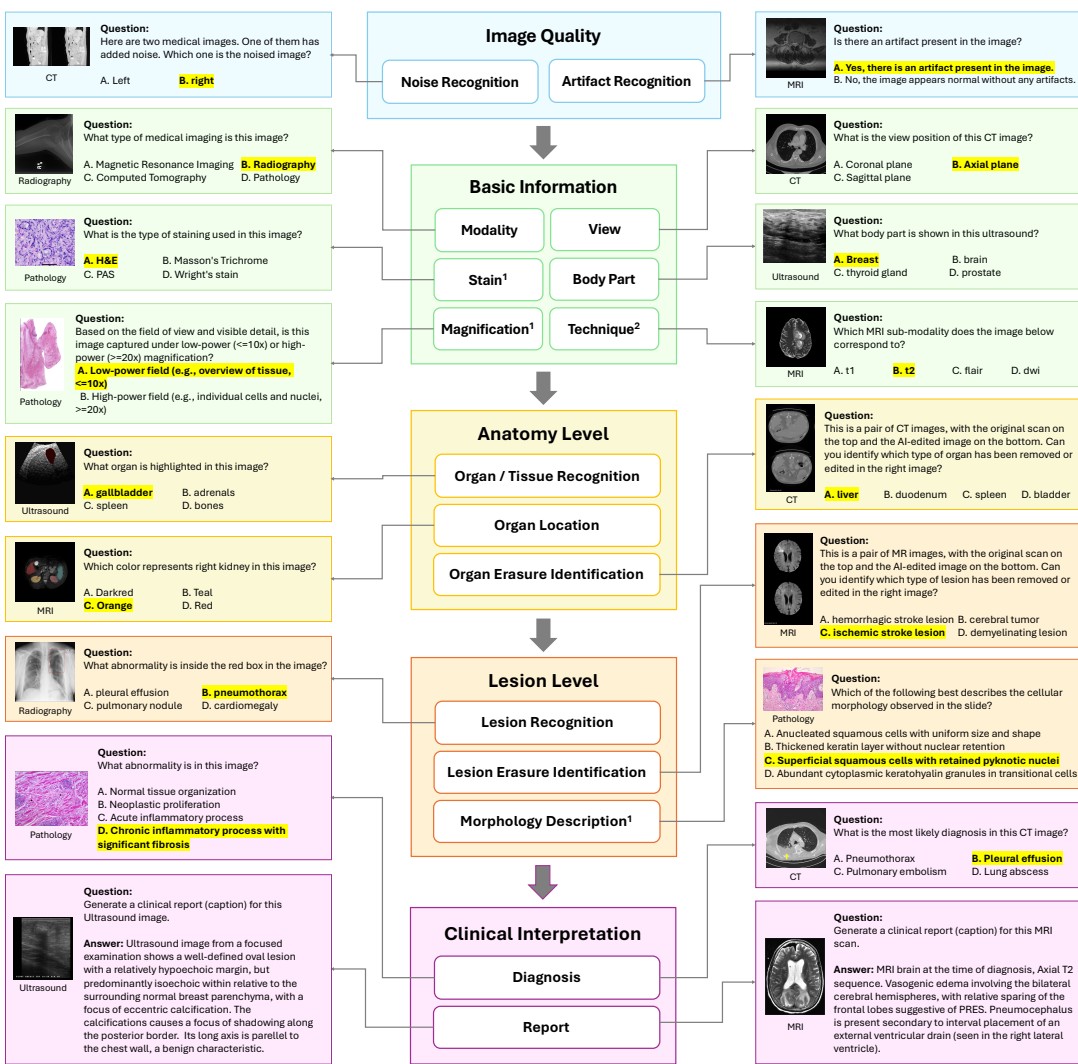

Figure 2: Hierarchical five-level evaluation framework for medical imaging diagnostics. Representative tasks—spanning CT, MRI, ultrasound, radiography, and pathology—are shown alongside their corresponding stage in the clinical reasoning cascade. Tasks labeled with [1] are exclusive to pathology, and those with [2] are exclusive to MRI.

dB—representing severe, moderate, and mild interference per clinical standards. Recognition and localization tasks include bounding box and region annotations for relevant anatomical or pathological areas. To assess sensitivity to local visual cues, we introduce an erasure task: using the Bailian high-throughput image platform, targeted anatomical or lesion regions are digitally removed while preserving structural continuity. All modified images are manually verified to ensure artifact-free erasure and reliable evaluation of visual evidence dependence.

The *Reasoning Trajectory Assessment* module builds on the *Visual Evidence Comprehension* question format but varies in structure and prompting. Independent QA pairs one image with multiple questions at different reasoning levels. Joint QA bundles sub-questions from multiple levels into a single prompt for simultaneous response. Multi-turn QA sequentially passes earlier responses into higher-level questions, simulating stepwise reasoning (See Appendix Figure 12).

For *Report Generation Evaluation*, we frame the task as medical captioning, using curated ground-truth references from PubMedVision [15] and PathMMU [73] that provide detailed clinical explanations.

All QA pairs are authored by physicians based on task objectives (e.g., modality recognition, localization, diagnosis) and span both closed-form and open-ended formats. Each is blind-reviewed by a senior clinician to ensure clinical relevance, visual grounding, and diagnostic realism. Distractors are carefully curated to remain plausible, and all questions are self-contained, unambiguous, and visually answerable. Data distribution is summarized in Table 2.

# 4 Experiments

## 4.1 Experiment Setup

In this study, we evaluated a total of 19 models (Table 3), including general-purpose open-source models, proprietary models accessed via API calls, and fine-tuned medical vision models. The parameter sizes of open-source models range from 7B to 72B. All experiments were conducted under a standardized zero-shot evaluation framework using system prompts on 8×NVIDIA A100 GPUs (80GB each). This setup ensures consistent and fair comparisons across different model architectures while maintaining reliable performance benchmarking.

Table 2: Statistics of DrVD-Bench

| Category | Metric | Count |
|---|---|---|
| *Module 1: Visual Evidence Comprehension* | | |
| Total QA pairs | | 4,480 |
| Level 0: Image Quality | QA pairs | 591 |
| Level 1: Basic Information | QA pairs | 1,400 |
| Level 2: Anatomy | QA pairs | 1,151 |
| Level 3: Lesion | QA pairs | 890 |
| Level 4: Clinical Interpretation | QA pairs | 923 |
| *Module 2: Reasoning Trajectory Assessment* | | |
| Total images | | 487 |
| Independent QA | QA pairs | 2,347 |
| Joint QA | QA pairs | 487 |
| Multi-turn QA | QA pairs | 487 |
| CT/MRI/Radiography/US | images each | 100 |
| Pathology | images | 87 |
| *Module 3: Report Generation Evaluation* | | |
| Open-ended QA pairs | | 475 |
| *Global Dataset Statistics* | | |
| Total QA pairs (all modules) | | 7,789 |
| Organ/Tissue classes | | 38 |
| Lesion classes | | 27 |
| Diagnosis categories | | 17 |

Table 3: VLMs benchmarked in our study

| Model | Developer | Year |
|---|---|---|
| *Proprietary* | | |
| GPT-4o[54] | OpenAI | 2024.11 |
| GPT-o1[55] | OpenAI | 2024.12 |
| GPT-o3[53] | OpenAI | 2025.04 |
| Gemini 2.5 Pro[25] | Google | 2025.03 |
| Grok-3[78] | xAI | 2025.02 |
| Doubao1.5-VisionPro [11] | ByteDance | 2025.01 |
| Claude 3.7 Sonnet[4] | Anthropic | 2025.02 |
| *Open-source* | | |
| Qwen2.5-VL [6] | Alibaba | 2025.01 |
| Phi-4 14B [1] | Microsoft | 2024.12 |
| GLM-4V [23] | Tsinghua | 2024.06 |
| Janus-Pro-7B [19] | DeepSeek | 2025.01 |
| *Medical-specific* | | |
| HuatuoGPT-Vision-34B [15] | CUHK | 2024.06 |
| HealthGPT-L14B [42] | ZJU | 2025.02 |
| RadFM-14B [76] | PJLAB | 2023.12 |
| LLaVA-Med-7B [39] | Microsoft | 2024.04 |

## 4.2 Evaluation

In DrVD-Bench, we use accuracy to evaluate multiple-choice tasks, each with a single correct answer. If the model's output conforms to the expected format, we directly compare it with the ground truth to determine correctness. For responses that deviate from the format, we apply DeepSeek-V3 [43] to extract the selected option. If extraction fails, the answer is marked as incorrect (See Appendix D). All results are averaged over five independent runs to ensure robustness. Accuracy is computed per question and then averaged within each level to obtain level-specific scores.

For open-ended tasks such as report generation, we employ DeepSeek-V3 [43] to extract key features from both the model's response and the reference text. We then adopt BERTScore [91] with PubMedBERT [26] to capture biomedical semantics of the generated data. To enable consistent comparison, we normalize BERTScore using the baseline (See Appendix B) and best-performing model. Let $s_{model}$ be the model's BERTScore, $s_{baseline}$ the score of an irrelevant response, and $s_{best}$

the highest score observed. The normalized score is:

$$\text{Normalized BERTScore} = \frac{s_{\text{model}} - s_{\text{baseline}}}{s_{\text{best}} - s_{\text{baseline}}}$$

In addition, to further evaluate factual consistency and hallucination in report generation, we incorporate FActScore [50] as a complementary metric. FActScore first extracts atomic-level clinical facts from both the reference and generated reports using DeepSeek-v3, followed by fact-level matching with GPT-4o. We compute two indicators: recall, which measures the proportion of true facts correctly captured by the model, and specificity, which measures the proportion of generated facts supported by the reference. Low recall and specificity respectively reflect insufficient factual coverage and the presence of hallucinated content.

## 5 Results Analysis

### 5.1 Models' Performance on Visual Evidence Comprehension

#### 5.1.1 Models Perform Differently Across Task Levels

Table 4 presents the evaluation results for CT modality tasks, with results for other modalities provided in Appendix B. Across all models, we observe a consistent decline in performance as the reasoning level increases. While most models perform well on basic recognition tasks such as identifying the imaging modality or view, their accuracy drops markedly on tasks requiring organ-level understanding, and declines even further on lesion-level reasoning. For instance, GPT-o3 achieves 86% accuracy on Basic Information tasks, decreases to 66% on Organ-Level tasks, and further falls to 41% on Lesion-Level tasks. This trend reflects a clear gap between surface-level visual parsing and clinically meaningful reasoning. As tasks increasingly demand multi-step inference and integration of both global context and local features, model performance becomes less stable and less reliable. These results highlight the limitations of current VLMs in replicating the fine-grained, layered reasoning processes central to clinical diagnosis.

Table 4: **Accuracy of different VLMs across different task levels.** The best-performing scores are highlighted in **red**, and the second-best in **blue**. Due to the large number of tasks across imaging modalities, we only present the performance of different models on task levels here. Results for the detailed subtasks among five modalities (CT, MRI, Ultrasound, Radiography, and Pathology) are provided in Appendix B. These are recognition subtasks across reasoning levels.

| Model | Image Quality | Basic Info | Anatomy Level | Lesion Level | Diagnosis |
|---|---|---|---|---|---|
| Random | 49 | 29 | 27 | 25 | 24 |
| **Proprietary** | | | | | |
| GPT-4o | 68 | 84 | 57 | 50 | 54 |
| GPT-o1 | 56 | 71 | 44 | 37 | 39 |
| GPT-o3 | 69 | 86 | 66 | 41 | 48 |
| Claude 3.7 Sonnet | 68 | 83 | 60 | 42 | 48 |
| Gemini 2.5 Pro | 76 | 88 | 65 | 52 | 54 |
| Grok-3 | 63 | 78 | 56 | 45 | 51 |
| Doubao-VisionPro | 63 | 82 | 52 | 59 | 52 |
| Qwen-VL-MAX | 65 | 78 | 54 | 56 | 53 |
| **Open-source** | | | | | |
| Qwen2.5-VL-72B | 65 | 77 | 54 | 56 | 52 |
| LLaVA-1.6-34B | 61 | 60 | 38 | 49 | 46 |
| Qwen2.5-VL-32B | 61 | 73 | 51 | 48 | 56 |
| Phi-4-14B | 70 | 68 | 39 | 44 | 47 |
| GLM-4V-9B | 65 | 70 | 43 | 32 | 36 |
| Qwen2.5-VL-7B | 68 | 69 | 41 | 45 | 38 |
| Janus-Pro-7B | 59 | 68 | 44 | 39 | 56 |
| **Medical-specific** | | | | | |
| HuaTuoGPT-Vision-34B | 61 | 85 | 58 | 54 | 59 |
| HealthGPT-L14B | 56 | 77 | 46 | 41 | 53 |
| RadFM-14B | 52 | 61 | 33 | 38 | 31 |
| LLaVA-Med-7B | 52 | 49 | 34 | 32 | 29 |

### 5.1.2 Overdiagnosis without Understanding

Notably, as shown in Table 4, many models perform better on diagnosis tasks than on lesion recognition, revealing a disconnect between output accuracy and reasoning fidelity. In other words, models can produce clinically plausible diagnostic results without actually identifying the supporting lesion evidence, a phenomenon we term "overdiagnosis without understanding".

This gap likely stems from biases in training data. Most VLMs are trained on image–report pairs that provide final diagnoses (e.g., "pneumonia", "fracture") but omit intermediate steps such as lesion localization or characterization. As a result, models tend to learn global pattern-to-label mappings, bypassing the step-by-step reasoning process that underpins clinical decision-making.

The issue becomes especially pronounced in zero-shot settings that demand fine-grained lesion-level reasoning. As shown in Table 11 (Appendix B), in CT lesion erasure detection, GPT-o3 and Gemini 2.5 Pro perform worse than random (19% and 16%, respectively), and achieve only 28% accuracy on lesion-level tasks—indicating a failure to recognize missing lesion evidence. Yet, on CT diagnosis tasks, they attain substantially higher accuracy (52% and 71%, respectively). This stark discrepancy suggests that diagnostic conclusions can be produced without properly grounding in anatomically or lesion-relevant visual features.

This behavior raises fundamental concerns about the reliability and clinical validity of current models. Despite producing seemingly accurate outputs, many models struggle with evidence-based reasoning, especially when visual grounding is essential to safe and explainable diagnosis.

## 5.2 Models Prefer Global Context over Step-by-Step Clinical Reasoning

Our benchmark reveals a clear performance hierarchy across the three reasoning settings (Table 5): Joint QA achieves the best overall results, followed by Independent QA, while Multi-turn QA performs the worst. In the Joint QA setting, where the model receives the full sequence of questions at once, it can reason more effectively by integrating global context and avoiding cumulative errors. Independent QA, where each question is asked separately without memory of previous turns, provides more stable but fragmented reasoning, leading to limited performance in higher-order tasks such as lesion identification and diagnosis. Surprisingly, Multi-turn QA—which retains previous questions and answers to simulate a realistic step-by-step clinical reasoning process—results in the weakest performance. This suggests that current models struggle with managing dialogue state and are vulnerable to propagating errors over turns. Overall, these findings indicate that today's VLMs benefit more from static, comprehensive context than from dynamic, trajectory-based reasoning.

To further verify this, we conducted a controlled experiment to isolate the effect of error accumulation from other factors such as missing image references or prompt formatting. We designed three conditions:

(1) **Baseline** — the model reasons based on its own previous answers;

(2) **Insert-Correct** — lower-level answers are replaced with ground-truth labels;

(3) **Insert-Error** — lower-level answers are replaced with random incorrect labels.

As shown in Table 6, results on three representative models (Qwen2.5-VL-72B, Claude 3.7 Sonnet, and GPT-4o) reveal that diagnosis accuracy improves notably when earlier answers are corrected (e.g., +28 for Qwen) and drops when errors are inserted (–15). Intermediate tasks (Body Part and Organ) change only slightly (±6%), suggesting that dependencies intensify at deeper reasoning stages. These results confirm that cumulative errors—rather than prompt design or forgetting—are the main cause of performance degradation in multi-turn reasoning.

## 5.3 Hallucinations and Reasoning Errors in Clinical Report Generation

Figure 9 and Table 17 (Appendix B) shows the report scores of various models across five imaging modalities. Gemini 2.5 Pro, HuatuoGPT, and Claude 3.7 Sonnet demonstrate strong performance across multiple modalities, with leading overall scores. Figure 3 further illustrates the performance of the highest-scoring model, Gemini 2.5 Pro, under different scenarios: examples A and B are high-scoring cases that successfully identify key lesions and generate structured descriptions, but still exhibit occasional diagnostic errors or hallucinations. In contrast, example C is a low-scoring

Table 5: **Performance across modality, bodypart, organ, lesion, and diagnosis (recognition subtasks across levels in module 1).** Each cell shows accuracy in the format: **Independent/Multi-turn/Joint**. The highest value(s) are **bolded**, second highest are underlined. Missing values are shown as –.

| Model | Modality | Bodypart | Organ | Lesion | Diagnosis |
|---|---|---|---|---|---|
| GPT-4o | **99/99/99** | **86**/85/– | 58/55/65 | 45/45/54 | 40/40/**41** |
| Claude | **99/99/99** | 82/**83**/– | 53/52/70 | 38/36/51 | 33/30/**38** |
| Gemini 2.5 Pro | **100/100**/99 | **90**/78/– | **62**/47/**76** | **51**/38/**61** | **48**/35/**55** |
| Qwen2.5-VL-72B | **99/99/99** | 85/**85**/– | 57/56/68 | 44/47/47 | 36/34/**38** |
| Grok-3 | **99/99/99** | **87**/59/– | 56/37/62 | 38/20/46 | 33/18/**35** |

Table 6: **Impact of low-level answer accuracy on multi-turn QA performance.** Each cell shows accuracy under three conditions: **Baseline/Insert-Correct/Insert-Error**. The highest value(s) are **bolded**, and the second highest are underlined.

| Model | Modality | Body Part | Organ | Lesion | Diagnosis |
|---|---|---|---|---|---|
| Qwen2.5-VL-72B | **99/99/99** | **85**/84/84 | 57/**58**/55 | 47/**60**/43 | 37/**65**/22 |
| Claude 3.7 Sonnet | **99/99/99** | 83/**84**/77 | 52/52/46 | 36/**50**/32 | 30/**60**/19 |
| GPT-4o | **99/99/99** | 85/**86**/85 | 55/55/54 | 45/**52**/44 | 40/**67**/33 |

case, where most of the generated content consists of hallucinated findings unrelated to the reference diagnosis. These results suggest that while current models have made noticeable progress in medical image-based reasoning and diagnosis, they still struggle to fully eliminate inaccuracies and hallucinations during the reasoning process, limiting their clinical reliability.

To quantify this issue, we evaluate four representative models—Gemini 2.5 Pro, HuaTuoGPT-Vision-34B, Qwen2.5-VL-72B, and Claude 3.7 Sonnet—using FActScore. As shown in Table 7, all models exhibit low recall and low specificity, indicating that they capture only a small fraction of clinically relevant facts while generating substantial hallucinated content unrelated to the reference reports or underlying images. These findings further demonstrate that factual consistency remains a key bottleneck for current VLMs in clinical report generation.

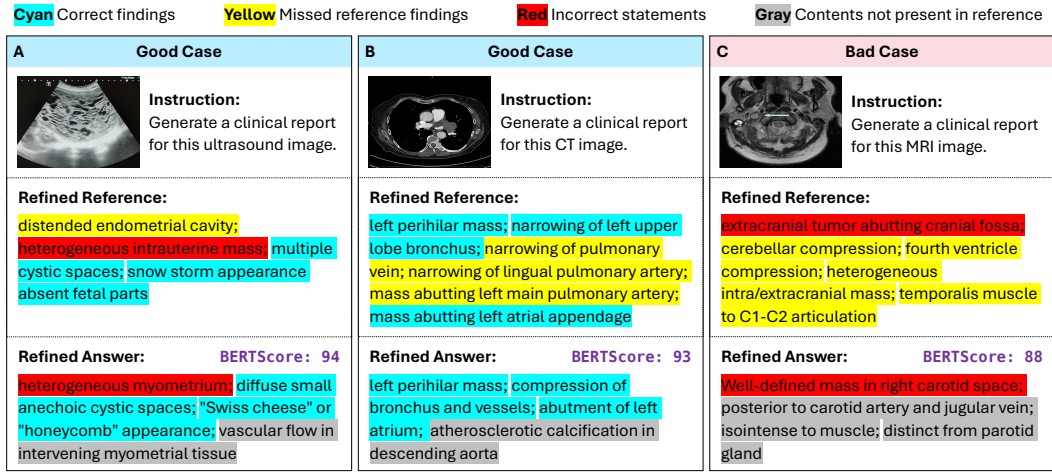

Figure 3: Report generation examples of Gemini 2.5 Pro. A and B represent high-scoring examples, while C represents bad-scoring examples

## 5.4 Bigger, Newer, or Specialized: Factors Drive Model Performance

Model performance on DrVD-Bench is shaped by both scale and recency. Large proprietary models such as Gemini 2.5 Pro and GPT-o3 lead the overall leaderboard, confirming the advantages of

Table 7: **FActScore results: factual consistency of clinical report generation.** Recall measures the coverage of true facts, while specificity reflects the proportion of generated facts supported by the reference report. Low values indicate persistent hallucination issues across models.

| Model | Recall (std) | Specificity (std) |
|---|---|---|
| Gemini 2.5 Pro | 0.13 (0.16) | 0.07 (0.10) |
| Claude 3.7 Sonnet | 0.06 (0.11) | 0.03 (0.05) |
| Qwen2.5-VL-7B | 0.08 (0.13) | 0.05 (0.09) |
| HuatuoGPT-Vision-34B | 0.13 (0.16) | 0.09 (0.17) |

scale—especially for complex reasoning and fine-grained visual tasks. Meanwhile, newer models consistently outperform older ones at similar sizes (Figure 4), highlighting the impact of improved architectures, training pipelines, and data quality on overall effectiveness.

Yet scale and recency are not the whole story. While proprietary models dominate, open-source models like Qwen2.5-VL-72B perform competitively despite having fewer parameters. More notably, HuaTuoGPT-Vision-34B achieves the second-highest accuracy with fewer than half the parameters of top-tier models. Its success demonstrates that domain-specific optimization—when aligned with medical structure and semantics—can enable smaller models to rival or even surpass larger general-purpose systems in clinical reasoning.

## 6    Conclusion

We present DrVD-Bench, a hierarchical, multimodal benchmark designed to assess whether VLMs reason like human clinicians. Covering five imaging modalities, 20 task types, and 7,789 QA pairs, DrVD-Bench captures the full spectrum of clinical visual reasoning—from basic information tasks like modality recognition to lesion identification and diagnosis. While many VLMs perform well on surface-level recognition tasks, their accuracy drops sharply as reasoning complexity increases. Notably, some models generate plausible diagnoses without correctly identifying supporting visual evidence, revealing a disconnect between diagnostic output and evidence-based understanding. By explicitly targeting intermediate reasoning steps and simulating clinical workflows, DrVD-Bench shows that current VLMs show early signs of clinical reasoning, but still far from truly interpreting medical images like human doctors. Our benchmark offers a structured foundation for developing clinically reliable and visually grounded medical AI systems.

## 7    Limitations

Although DrVD-Bench is clinically inspired, it remains a controlled evaluation without real-world patient context (e.g., notes, disease progression), limiting its reflection of actual diagnostic workflows. Thus, results should not be overinterpreted for clinical translation, and future refinements should better capture real-world complexity.

As DrVD-Bench uses public datasets, some recent models may have seen parts of them during pre-training, leading to potential train–test overlap and affecting performance trends. Besides, given current VLM limitations, selecting one "most informative" 2D slice from 3D or 4D volumes is a practical compromise but remains a common constraint. Moreover, most training datasets (Table 10 in Appendix A) originate from large institutions,

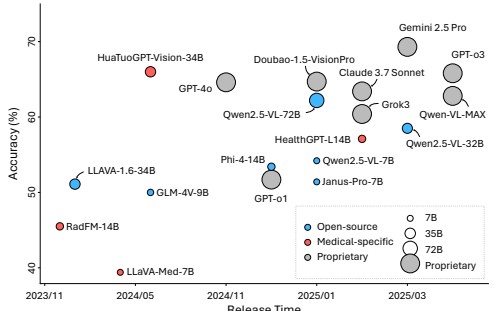

Figure 4: Performance of VLMs on DrVD-Bench visual evidence comprehension tasks across different scales and recencies.

underrepresenting minority groups and smaller or resource-limited hospitals. Future benchmarks should better control data leakage, enable volumetric reasoning, and reduce data bias to improve fairness and generalizability.

## Acknowledgement

This work was supported by the National Natural Science Foundation of China (92470105, 62373210), the National Key R&D Program of China (2025YFC3409300), and the Tsinghua-Toyota Joint Research Institute Inter-disciplinary Program (20243930093).

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

# A  Technical Appendices and Supplementary Material

## A.1  Dataset Availability

**DrVD-Bench** is publicly available at the following links:

- **Kaggle**: `https://www.kaggle.com/datasets/tianhongzhou/drvd-bench`
- **Hugging Face**: `https://huggingface.co/datasets/jerry1565/DrVD-Bench`

The code used to conduct our experiments is released at `https://github.com/Jerry-Boss/DrVD-Bench`.

## A.2  Detailed Composition of Our Benchmark

In this section, we describe the task compositions and dataset usage in more detail. Our benchmark contains 3 modules, Module 1 (Visual Evidence Comprehension) contains 4480 QA pairs, Module 2 (Reasoning Trajectory Assessment) contains 3321 QAs, and Module 3 (Report Generation Evaluation) contains 475 questions (See Figure 5). The compositions of Module 1 is illustrated in Figure 6, and that of Module 2 is illustrated in Figure 7. The levels and tasks of Module 1 are listed in detail in Table 9.

Tasks of Module 2 (Reasoning Trajectory Assessment) are adapted from Module 1 (Visual Evidence Comprehension), But they are differently organized. They are organized into 3 forms using different prompts (See Appendix C). Notably, there are 4 questions per Joint QA, and 5 questions per Multi-turn QA.

In Joint QA, the number of answer options is carefully designed to reduce prompt-induced information leakage. Both modality and organ questions contain 4 options each, while lesion and diagnosis questions are expanded to 8 options (i.e., 4×2), mitigating the information leak from our prompts.

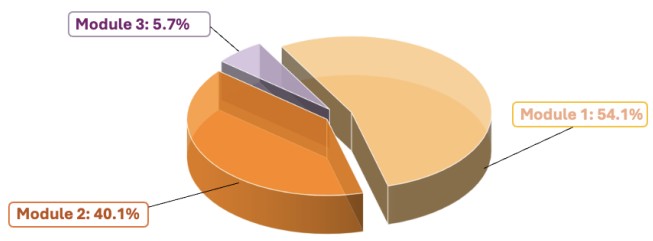

Figure 5: Diagram for the composition of our benchmark, which contains 3 modules

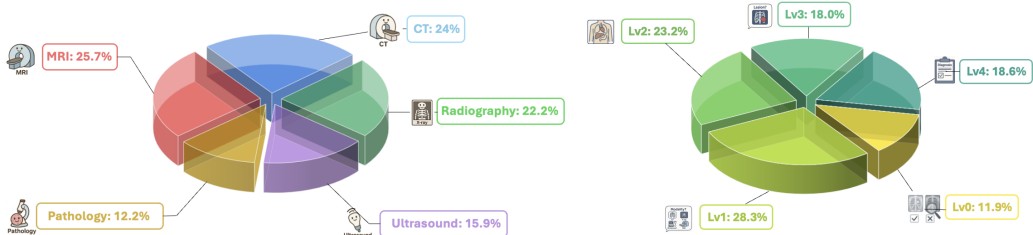

Figure 6: Diagram for the composition of Module 1. Module 1 spans all five modalities, and is designed to contain tasks of five levels

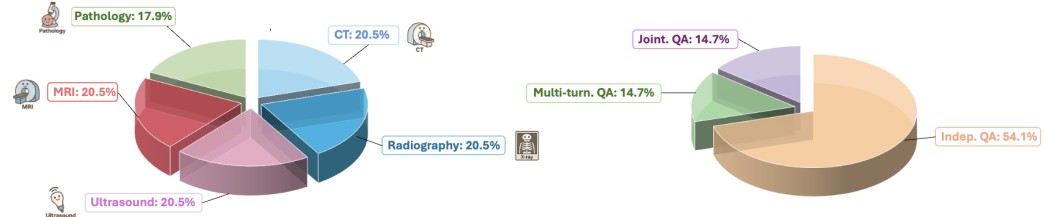

Figure 7: Diagram for the composition of Module 2. Module 2 contains three parts: Independent QA, Joint QA and Multi-turn QA.

Table 8: Datasets used in DrVD-Bench, organized by imaging modality.

| Modality | Dataset Name |
|---|---|
| **Radiography (X-ray)** | MIMIC-CXR v4[34] |
| | MURA[64] |
| | VinDr-SpineXR [52] |
| | Panoramic Dental Radiography / Tufts Dental Dataset[56] |
| | SA-Med2D-20M [84] |
| | RSNA-Pneumonia[3] |
| **CT** | SA-Med2D-20M [84] |
| | AMOS 2022 [33] |
| | DeepLesion[81] |
| | CT-RATE[28] |
| | PubMedVision [15] |
| | LiTS (Liver Tumor Segmentation)[10] |
| | COVID-CTset [63] |
| | CTPelvic1k [46] |
| | CT-ORG [65] |
| | StructSeg2019-subtask1 [70] |
| | KiTS 2021 [30] |
| | LNDb [57] |
| | MSD-Liver [5, 71] COVID-19-20 [67] |
| **MRI** | TotalSegmentator MRI[21] |
| | BraTS 2020[49, 7, 8] |
| | BraTS 2021 [49, 7, 8] |
| | PI-CAI [69] |
| | LLD-MMRI [48] |
| | MICCAI 2024 CARE MyoPS++ [94, 40, 61, 22] |
| | ISPY1-Tumor-SEG-Radiomics[20] |
| **Ultrasound** | BUSI[2] |
| | CardiacUDA[83] |
| | CuRIOUS2022[80, 9] |
| | TG3K [24] |
| | Abdominal Ultrasound Images [12] |
| | Annotated Ultrasound Liver Images [86] |
| **Pathology** | PathMMU[73] |

Table 9: Task QA numbers in Module 1 (continued on next page)

| Modality | Level | Task | Number |
|---|---|---|---|
| CT | Image Quality | Noise Recognition | 100 |
| | | Artifact Recognition | 100 |
| | Basic Information | Modality | 100 |
| | | View | 100 |
| | | Body Part | 100 |
| | Anatomy Level | Organ Recognition | 100 |
| | | Organ Location | 99 |
| | | Organ Erasure Identification | 98 |
| | Lesion Level | Lesion Recognition | 95 |
| | | Lesion Erasure Identification | 100 |
| | Clinical Interpretation | Report | 100 |
| | | Diagnosis | 90 |
| Radiography | Image Quality | Noise Recognition | 100 |
| | | Artifact Recognition | 61 |
| | Basic Information | Modality | 100 |
| | | View | 100 |
| | | Body Part | 100 |
| | Anatomy Level | Organ Recognition | 100 |
| | | Organ Location | 100 |
| | | Organ Erasure Identification | 68 |
| | Lesion Level | Lesion Recognition | 95 |
| | | Lesion Erasure Identification | 100 |
| | Clinical Interpretation | Report | 100 |
| | | Diagnosis | 100 |
| Ultrasound | Image Quality | Artifact Recognition | 46 |
| | Basic Information | Modality | 100 |
| | | Body Part | 100 |
| | Anatomy Level | Organ Recognition | 100 |
| | | Organ Erasure Identification | 100 |
| | Lesion Level | Lesion Recognition | 100 |
| | | Lesion Erasure Identification | 100 |
| | Clinical Interpretation | Report | 75 |
| | | Diagnosis | 63 |
| Pathology | Basic Information | Stain | 100 |
| | | Magnification | 100 |
| | Anatomy Level | Organ/Tissue Recognition | 100 |
| | Lesion Level | Morphology Description | 100 |
| | Clinical Interpretation | Report | 100 |
| | | Diagnosis | 100 |

| Modality | Level | Task | Number |
|---|---|---|---|
| MRI | Image Quality | Noise Recognition | 100 |
| | | Artifact Recognition | 84 |
| | Basic Information | Modality | 100 |
| | | View | 100 |
| | | Technique | 100 |
| | | Body Part | 100 |
| | Anatomy Level | Organ Recognition | 100 |
| | | Organ Location | 86 |
| | | Organ Erasure Identification | 100 |
| | Lesion Level | Lesion Recognition | 100 |
| | | Lesion Erasure Identification | 100 |
| | Clinical Interpretation | Report | 100 |
| | | Diagnosis | 95 |

Table 10: Training datasets for medical-Specific VLMs. List of domain-specific vision–language models and their corresponding medical training datasets.

| Medical-specific VLM | Medical training datasets |
|---|---|
| HuatuoGPT-Vision-34B | PubMedVision |
| HealthGPT-L14B | PubMedVision, LLaVA-Med, PathVQA, MIMIC-CXR-VQA, SLAKE, VQA-RAD |
| RadFM-14B | Rad3D-series, MPx-series, PMC-Inline, PMC-CaseReport, VinDr-Mammo, VinDr-SpineXR, VinDr-PCXR, PMC-OA, PMC-VQA, VQA-RAD, SLAKE, MIMIC-CXR, VinDr-CXR, NIH ChestXray14, CheXpert, Covid-CXR2, NLM-TB, Object-CXR, OpenI |
| LLaVA-Med-7B | PMC-15M |

# B   Detailed Results

In this section, we show our benchmark results in detail.

For Module 1 (Visual Evidence Comprehension), the result for CT is in Table 11; the result for Radiography is in Table 12; the result for MRI is in Table 13; the result for Ultrasound is in Table 14; the result for Pathology is in Table 15.

We also analyzed model performance across different imaging modalities (Figure 8). VLMs perform best on Pathology and worst on Ultrasound. The strong results in Pathology may stem from the fact that we did not use whole-slide images; instead, human annotators selected diagnostically relevant regions and zoomed in, effectively reducing task complexity. In contrast, Ultrasound poses a unique challenge due to its inherently dynamic nature—clinical interpretation typically relies on real-time video sequences rather than static frames, making single-image reasoning considerably more difficult.

We evaluated the performance of different models on the report generation task by comparing their outputs against clinically relevant references (See Figure 9 and Table 17). The performance differences across modalities is minimal.

To establish a baseline, we introduced a set of medically plausible but image-irrelevant texts and computed their BERTScores against the ground-truth references. Each reference was paired with a randomly selected sentence from the list below, and the resulting BERTScore served as the normalization baseline:

- No focal consolidation is seen. However, based on clinical history, the findings may suggest a prior episode of viral gastroenteritis.

- There is no acute intracranial hemorrhage. The patient's recent weight loss should be evaluated further with laboratory studies.

- No pulmonary embolism is identified. The patient's chronic insomnia is unlikely to be explained by these imaging findings.

- Normal appearance of abdominal organs. Note: elevated serum calcium should be correlated with parathyroid hormone levels.

- No significant degenerative changes are observed. Patient's dizziness may be related to recent changes in medication dosage.

- Imaging of the chest is unremarkable. Recommend thyroid function tests given the history of fatigue and cold intolerance.

- No mass lesion is detected. Given the positive ANA, autoimmune evaluation is advised.

- The study is negative for acute pathology. Further investigation is warranted for the reported night sweats and low-grade fever.

- No obstructive uropathy is evident. Patient's lab findings of hematuria require correlation with urine cytology.

- Brain MRI is within normal limits. Symptoms of memory loss may be functional in origin or related to recent psychosocial stressors.

Table 11: **Results of different VLMs across different VQA tasks in visual evidence tasks for the CT modality.** The best-performing scores are bolded, and the second-best are underlined.

| Model | Image Quality | | | Basic Information | | | | Organ Level | | | | Lesion Level | | | Clinical Interpretation |
|---|---|---|---|---|---|---|---|---|---|---|---|---|---|---|---|
| | Overall | Artifact | Noise | Overall | Modality | Bodypart | View | Overall | Recognition | Location | Erasure | Overall | Recognition | Erasure | Diagnosis |
| Random | 48 | 51 | 45 | 30 | 25 | 27 | 39 | 29 | 29 | 27 | 30 | 26 | 25 | 26 | 24 |
| Proprietary | | | | | | | | | | | | | | | |
| GPT-4o | 72 | 52 | 92 | 89 | 100 | 85 | 92 | 50 | 60 | 46 | 44 | 38 | 39 | 36 | 67 |
| GPT-o1 | 62 | 51 | 72 | 86 | 94 | 78 | 87 | 35 | 39 | 34 | 33 | 29 | 30 | 27 | 46 |
| GPT-o3 | 80 | 65 | 94 | 94 | 100 | 90 | 92 | 70 | 76 | 77 | 56 | 28 | 37 | 19 | 52 |
| Claude 3.7 Sonnet | 75 | 52 | 97 | 84 | 100 | 84 | 92 | 47 | 60 | 46 | 36 | 29 | 28 | 29 | 46 |
| Gemini 2.5 Pro | 82 | 64 | 99 | 93 | 100 | 88 | 92 | 67 | 67 | 73 | 62 | 28 | 39 | 16 | 71 |
| Grok-3 | 65 | 54 | 76 | 66 | 100 | 88 | 43 | 44 | 58 | 33 | 41 | 35 | 36 | 33 | 58 |
| Doubao1.5-VisionPro | 78 | 68 | 87 | 90 | 98 | 81 | 90 | 50 | 59 | 52 | 38 | 55 | 61 | 49 | 59 |
| Qwen-VL-MAX | 74 | 61 | 87 | 87 | 100 | 85 | 89 | 54 | 55 | 58 | 49 | 45 | 42 | 47 | 68 |
| Open-source | | | | | | | | | | | | | | | |
| Qwen2.5-VL-72B | 75 | 59 | 90 | 88 | 100 | 87 | 89 | 54 | 57 | 61 | 45 | 45 | 42 | 47 | 69 |
| LLaVA-1.6-34B | 62 | 53 | 70 | 72 | 80 | 85 | 51 | 32 | 42 | 16 | 38 | 43 | 41 | 45 | 54 |
| Qwen2.5-VL-32B | 59 | 50 | 68 | 88 | 97 | 80 | 86 | 45 | 48 | 41 | 47 | 39 | 46 | 32 | 68 |
| Phi-4-14B | 75 | 50 | 99 | 67 | 99 | 76 | 57 | 36 | 45 | 27 | 36 | 41 | 35 | 46 | 64 |
| GLM-4V-9B | 66 | 49 | 83 | 74 | 86 | 83 | 52 | 36 | 44 | 27 | 36 | 33 | 29 | 36 | 38 |
| Qwen2.5-VL-7B | 71 | 57 | 84 | 76 | 100 | 79 | 73 | 37 | 40 | 34 | 36 | 40 | 42 | 38 | 39 |
| Janus-Pro-7B | 71 | 52 | 90 | 82 | 94 | 88 | 63 | 37 | 45 | 32 | 34 | 41 | 40 | 42 | 73 |
| Medical-specific | | | | | | | | | | | | | | | |
| HuaTuoGPT-Vision-34B | 67 | 60 | 74 | 94 | 100 | 91 | 92 | 52 | 63 | 48 | 44 | 33 | 47 | 33 | 68 |
| HealthGPT-L14B | 51 | 49 | 53 | 84 | 100 | 90 | 84 | 38 | 52 | 30 | 32 | 32 | 41 | 22 | 72 |
| RadFM-14B | 52 | 47 | 56 | 72 | 92 | 56 | 67 | 30 | 39 | 29 | 22 | 44 | 45 | 42 | 35 |
| LLaVA-Med-7B | 57 | 53 | 60 | 55 | 82 | 57 | 26 | 33 | 32 | 36 | 32 | 34 | 33 | 35 | 30 |

Table 12: **Results of different VLMs across different VQA tasks in visual evidence tasks for the radiography modality.** The best-performing scores are **bolded**, and the second-best are underlined.

| Model | Image Quality | | | Basic Information | | | | Organ Level | | | | Lesion Level | | | Clinical Interpretation |
|---|---|---|---|---|---|---|---|---|---|---|---|---|---|---|---|
| | Overall | Artifact | Noise | Overall | Modality | Bodypart | View | Overall | Recognition | Location | Erasure | Overall | Recognition | Erasure | Diagnosis |
| Random | 49 | 51 | 46 | 27 | 29 | 25 | 28 | 30 | 18 | 44 | 27 | 25 | 28 | 22 | 21 |
| Proprietary | | | | | | | | | | | | | | | |
| GPT-4o | 69 | 70 | 67 | 92 | 100 | 95 | 81 | 59 | 46 | 79 | 53 | 36 | 45 | 27 | 27 |
| GPT-o1 | 60 | 61 | 59 | 82 | 91 | 74 | 81 | 51 | 49 | 57 | 48 | 33 | 40 | 25 | 18 |
| GPT-o3 | 72 | 74 | 69 | 92 | 100 | 98 | 79 | 76 | 80 | 94 | 53 | 33 | 41 | 25 | 24 |
| Claude 3.7 Sonnet | 62 | 62 | 61 | 95 | 100 | 91 | 93 | 66 | 81 | 65 | 53 | 38 | 47 | 28 | 36 |
| Gemini 2.5 Pro | 79 | 70 | 88 | 88 | 100 | 97 | 68 | 71 | 80 | 89 | 43 | 53 | 57 | 48 | 25 |
| Grok-3 | 64 | 72 | 55 | 91 | 99 | 85 | 90 | 55 | 37 | 74 | 53 | 32 | 36 | 28 | 32 |
| Doubao-VisionPro | 53 | 44 | 61 | 90 | 100 | 89 | 82 | 60 | 84 | 77 | 18 | 37 | 43 | 30 | 42 |
| Qwen-VL-MAX | 66 | 68 | 64 | 85 | 100 | 93 | 63 | 71 | 80 | 78 | 56 | 37 | 37 | 36 | 27 |
| Open-source | | | | | | | | | | | | | | | |
| Qwen2.5-VL-72B | 65 | 63 | 67 | 76 | 100 | 63 | 64 | 70 | 79 | 78 | 52 | 41 | 48 | 34 | 28 |
| LLaVA-1.6-34B | 62 | 72 | 51 | 69 | 96 | 65 | 46 | 37 | 38 | 62 | 10 | 25 | 27 | 23 | 31 |
| Qwen2.5-VL-32B | 64 | 62 | 66 | 67 | 100 | 63 | 37 | 68 | 75 | 78 | 52 | 33 | 38 | 27 | 29 |
| Phi-4-14B | 77 | 74 | 84 | 75 | 100 | 90 | 34 | 43 | 39 | 73 | 16 | 31 | 34 | 27 | 37 |
| GLM-4V-9B | 66 | 64 | 68 | 81 | 100 | 89 | 53 | 50 | 56 | 66 | 28 | 28 | 29 | 26 | 20 |
| Qwen2.5-VL-7B | 75 | 78 | 72 | 73 | 98 | 89 | 33 | 46 | 53 | 61 | 25 | 33 | 36 | 30 | 37 |
| Janus-Pro-7B | 57 | 46 | 67 | 73 | 99 | 88 | 33 | 39 | 37 | 73 | 6 | 33 | 35 | 30 | 24 |
| Medical-specific | | | | | | | | | | | | | | | |
| HuaTuoGPT-Vision-34B | 63 | 66 | 59 | 81 | 100 | 63 | 79 | 58 | 67 | 75 | 32 | 35 | 42 | 28 | 37 |
| HealthGPT-L14B | 60 | 70 | 49 | 75 | 98 | 60 | 68 | 39 | 32 | 66 | 19 | 32 | 37 | 27 | 34 |
| RadFM-14B | 60 | 70 | 49 | 66 | 72 | 58 | 68 | 39 | 34 | 51 | 31 | 24 | 17 | 30 | 44 |
| LLaVA-Med-7B | 51 | 49 | 52 | 51 | 70 | 51 | 32 | 44 | 28 | 60 | 44 | 28 | 25 | 31 | 24 |

Table 13: **Results of different VLMs across different VQA tasks in visual evidence tasks for the MRI modality.** The best-performing scores are **bolded**, and the second-best are underlined.

| Model | Image Quality | | | Basic Information | | | | Organ Level | | | | Lesion Level | | | Clinical Interpretation |
|---|---|---|---|---|---|---|---|---|---|---|---|---|---|---|---|
| | Overall | Artifact | Noise | Overall | Modality | Bodypart | Imaging | Overall | Recognition | Location | Erasure | Overall | Recognition | Erasure | Diagnosis |
| Random | 49 | 55 | 42 | 27 | 21 | 29 | 24 | 34 | 24 | 22 | 29 | 20 | 28 | 25 | 30 |
| Proprietary | | | | | | | | | | | | | | | |
| GPT-4o | 66 | 51 | 80 | 72 | 100 | 68 | 55 | 66 | 52 | 66 | 45 | 46 | 58 | 53 | 57 |
| GPT-o1 | 50 | 43 | 67 | 56 | 78 | 46 | 36 | 65 | 35 | 38 | 29 | 38 | 44 | 46 | 40 |
| GPT-o3 | 62 | 40 | 84 | 76 | 100 | 69 | 62 | 71 | 53 | 66 | 49 | 43 | 51 | 50 | 48 |
| Claude 3.7 Sonnet | 70 | 54 | 86 | 65 | 100 | 56 | 43 | 60 | 49 | 56 | 38 | 53 | 56 | 61 | 43 |
| Gemini 2.5 Pro | 70 | 45 | 94 | 80 | 99 | 74 | 81 | 67 | 54 | 58 | 46 | 57 | 55 | 55 | 57 |
| Grok-3 | 60 | 45 | 75 | 64 | 98 | 65 | 37 | 55 | 51 | 64 | 35 | 53 | 50 | 51 | 50 |
| Doubao-VisionPro | 62 | 56 | 68 | 64 | 99 | 53 | 45 | 60 | 46 | 61 | 38 | 40 | 61 | 54 | 45 |
| Qwen-VL-MAX | 62 | 45 | 79 | 61 | 96 | 58 | 37 | 54 | 43 | 48 | 36 | 46 | 65 | 68 | 60 |
| Open-source | | | | | | | | | | | | | | | |
| Qwen2.5-VL-72B | 65 | 50 | 79 | 63 | 96 | 60 | 40 | 54 | 46 | 54 | 36 | 49 | 67 | 71 | 58 |
| LLaVA-1.6-34B | 62 | 50 | 74 | 46 | 57 | 43 | 29 | 56 | 35 | 36 | 24 | 45 | 68 | 60 | 76 |
| Qwen2.5-VL-32B | 60 | 49 | 71 | 62 | 91 | 60 | 39 | 57 | 44 | 45 | 38 | 48 | 63 | 61 | 70 |
| Phi-4-14B | 66 | 43 | 88 | 55 | 97 | 47 | 25 | 51 | 31 | 33 | 24 | 36 | 57 | 49 | 44 |
| GLM-4V-9B | 65 | 50 | 79 | 50 | 93 | 55 | 17 | 34 | 39 | 39 | 30 | 47 | 35 | 29 | 37 |
| Qwen2.5-VL-7B | 62 | 45 | 79 | 51 | 95 | 52 | 24 | 34 | 42 | 49 | 29 | 47 | 54 | 46 | 39 |
| Janus-Pro-7B | 59 | 38 | 79 | 57 | 87 | 68 | 22 | 50 | 43 | 45 | 44 | 40 | 36 | 31 | 79 |
| Medical-specific | | | | | | | | | | | | | | | |
| HuaTuoGPT-Vision-34B | 57 | 40 | 73 | 77 | 100 | 70 | 65 | 73 | 49 | 63 | 36 | 47 | 68 | 67 | 67 |
| HealthGPT-L14B | 60 | 67 | 53 | 61 | 98 | 63 | 25 | 58 | 43 | 50 | 35 | 43 | 54 | 54 | 42 |
| RadFM-14B | 46 | 50 | 42 | 49 | 94 | 31 | 38 | 33 | 43 | 58 | 32 | 40 | 55 | 50 | 35 |
| LLaVA-Med-7B | 51 | 59 | 43 | 43 | 71 | 34 | 28 | 38 | 25 | 31 | 17 | 27 | 36 | 34 | 32 |

Table 14: **Results of different VLMs across different VQA tasks in visual evidence tasks for the ultrasound modality.** The best-performing scores are **bolded**, and the second-best are underlined.

| Model | Image Quality | Basic Information | | | Organ Level | | | Lesion Level | | | Clinical Interpretation |
|---|---|---|---|---|---|---|---|---|---|---|---|
| | Artifact | Overall | Modality | Bodypart | Overall | Recognition | Erasure | Overall | Recognition | Erasure | Diagnosis |
| Random | 53 | 25 | 22 | 27 | 24 | 23 | 25 | 22 | 23 | 20 | 20 |
| *Proprietary* | | | | | | | | | | | |
| GPT-4o | 50 | 76 | 99 | 53 | 44 | 37 | 51 | 54 | 76 | 31 | 62 |
| GPT-o1 | 35 | 61 | 89 | 32 | 35 | 27 | 43 | 35 | 44 | 26 | 43 |
| GPT-o3 | 46 | **82** | 99 | **65** | 46 | 36 | 55 | 36 | 36 | 36 | 48 |
| Claude 3.7 Sonnet | 48 | 71 | **100** | 41 | 54 | **46** | 61 | 28 | 36 | 19 | 49 |
| Gemini 2.5 Pro | 50 | 81 | **100** | 61 | 48 | 40 | 56 | 59 | 58 | 59 | 51 |
| Grok-3 | **54** | 75 | 98 | 52 | 45 | 36 | 54 | 53 | 74 | 32 | **64** |
| Doubao-VisionPro | 48 | 71 | **100** | 42 | 35 | 31 | 39 | **83** | **87** | **78** | 49 |
| Qwen-VL-MAX | 39 | 67 | 98 | 35 | 37 | 33 | 40 | 69 | 76 | 62 | 52 |
| *Open-source* | | | | | | | | | | | |
| Qwen2.5-VL-72B | 30 | 66 | 98 | 33 | 36 | 33 | 38 | 63 | 74 | 52 | 46 |
| LLaVA-1.6-34B | 50 | 54 | 77 | 31 | 32 | 22 | 42 | 58 | 83 | 32 | 51 |
| Qwen2.5-VL-32B | 50 | 59 | 94 | 24 | 32 | 28 | 36 | 34 | 36 | 32 | 52 |
| Phi-4-14B | 37 | 66 | 95 | 36 | 26 | 25 | 27 | 43 | 49 | 36 | 41 |
| GLM-4V-9B | 50 | 75 | **100** | 49 | 29 | 30 | 28 | 17 | 23 | 10 | 33 |
| Qwen2.5-VL-7B | 50 | 47 | 92 | 2 | 30 | 32 | 28 | 46 | 36 | 56 | 25 |
| Janus-Pro-7B | 15 | 69 | 98 | 40 | 30 | 29 | 30 | 26 | 47 | 4 | 60 |
| *Medical-specific* | | | | | | | | | | | |
| HuaTuoGPT-Vision-34B | 46 | **82** | 98 | **65** | 44 | 33 | 55 | 64 | 63 | 64 | **64** |
| HealthGPT-L14B | 50 | 67 | 94 | 40 | 35 | 36 | 34 | 25 | 4 | 45 | 51 |
| RadFM-14B | 50 | 47 | 60 | 33 | 20 | 25 | 14 | 36 | 60 | 12 | 11 |
| LLaVA-Med-7B | 44 | 50 | 58 | 41 | 23 | 21 | 25 | 31 | 30 | 32 | 30 |

Table 15: **Results of different VLMs across different VQA tasks in visual evidence tasks for the pathology modality.** The best-performing scores are **bolded**, and the second-best are underlined.

| Model | Basic Information | | | Organ Level | Lesion Level | Clinical Interpretation |
|---|---|---|---|---|---|---|
| | Overall | Magnification | Stain | Recognition | Morphology | Diagnosis |
| Random | 34 | 46 | 21 | 30 | 23 | 25 |
| *Proprietary* | | | | | | |
| GPT-4o | 88 | 88 | 88 | 80 | 62 | 57 |
| GPT-o1 | 72 | 73 | 71 | 65 | 43 | 50 |
| GPT-o3 | 88 | 89 | 87 | 83 | 56 | 66 |
| Claude 3.7 Sonnet | 94 | 96 | 91 | 83 | 61 | **67** |
| Gemini 2.5 Pro | **96** | **99** | **93** | **86** | 63 | **67** |
| Grok-3 | 83 | 83 | 83 | 83 | 55 | 49 |
| Doubao-VisionPro | 93 | 94 | 91 | 69 | 58 | 63 |
| Qwen-VL-MAX | 86 | 84 | 87 | 65 | 63 | 58 |
| *Open-source* | | | | | | |
| Qwen2.5-VL-72B | 86 | 85 | 87 | 66 | 64 | 59 |
| LLaVA-1.6-34B | 60 | 56 | 63 | 53 | 51 | 49 |
| Qwen2.5-VL-32B | 88 | 89 | 87 | 66 | **69** | 59 |
| Phi-4-14B | 69 | 65 | 72 | 59 | 46 | 47 |
| GLM-4V-9B | 69 | 73 | 65 | 61 | 48 | 51 |
| Qwen2.5-VL-7B | 89 | 93 | 84 | 50 | 50 | 50 |
| Janus-Pro-7B | 58 | 70 | 46 | 73 | 57 | 44 |
| *Medical-specific* | | | | | | |
| HuaTuoGPT-Vision-34B | 93 | 93 | **93** | **86** | 61 | 57 |
| HealthGPT-L14B | 91 | 93 | 89 | 75 | 64 | 64 |
| RadFM-14B | 73 | 60 | 85 | 32 | 33 | 30 |
| LLaVA-Med-7B | 47 | 56 | 38 | 43 | 32 | 27 |

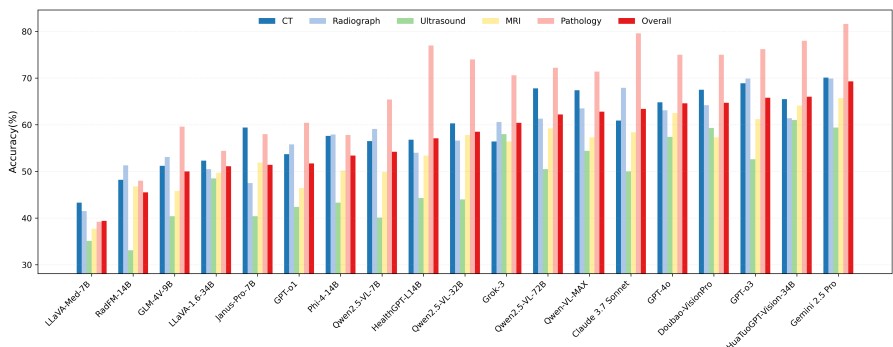

Figure 8: Performance of VLMs across five medical imaging modalities in visual evidence tasks, sorted by overall accuracy.

Table 16: **Performance of VLMs across five medical imaging modalities in visual evidence tasks.** The best-performing scores are **bolded**, and the second-best are underlined.

| Model | CT | Radiography | Ultrasound | MRI | Pathology | Overall |
|---|---|---|---|---|---|---|
| **Proprietary** | | | | | | |
| GPT-4o | 64.8 | 63.1 | 57.4 | 62.5 | 75 | 64.6 |
| GPT-o1 | 53.7 | 55.8 | 42.4 | 46.4 | 60.4 | 51.7 |
| GPT-o3 | 68.9 | **69.9** | 52.6 | 61.2 | 76.2 | 65.8 |
| Claude 3.7 Sonnet | 60.9 | 67.9 | 50 | 58.4 | 79.6 | 63.4 |
| Gemini 2.5 Pro | **70.1** | **69.9** | 59.4 | **65.7** | **81.6** | **69.3** |
| Grok-3 | 56.4 | 60.6 | 58 | 56.4 | 70.6 | 60.4 |
| Doubao-1.5-VisionPro | 67.5 | 64.2 | 59.3 | 57.3 | 75 | 64.7 |
| Qwen-VL-MAX | 67.4 | 63.5 | 54.4 | 57.3 | 71.4 | 62.8 |
| **Open-source** | | | | | | |
| Qwen2.5-VL-72B | 67.8 | 61.3 | 50.5 | 59.2 | 72.2 | 62.2 |
| LLaVA-1.6-34B | 52.3 | 50.5 | 48.5 | 49.7 | 54.4 | 51.1 |
| Qwen2.5-VL-32B | 60.3 | 56.6 | 44 | 57.8 | 74 | 58.5 |
| Phi-4-14B | 57.6 | 57.9 | 43.3 | 50.2 | 57.8 | 53.4 |
| GLM-4V-9B | 51.2 | 53.1 | 40.4 | 45.8 | 59.6 | 50.0 |
| Qwen2.5-VL-7B | 56.5 | 59.1 | 40.1 | 49.9 | 65.4 | 54.2 |
| Janus-Pro-7B | 59.4 | 47.5 | 40.4 | 51.9 | 58 | 51.4 |
| **Medical-specific** | | | | | | |
| HuaTuoGPT-Vision-34B | 65.5 | 61.4 | **61** | 64.1 | 78 | 66 |
| HealthGPT-L14B | 56.8 | 54 | 44.3 | 53.4 | 77 | 57.1 |
| RadFM-14B | 48.2 | 51.3 | 33.1 | 46.8 | 48 | 45.5 |
| LLaVA-Med-7B | 43.3 | 41.5 | 35.1 | 37.7 | 39.2 | 39.4 |

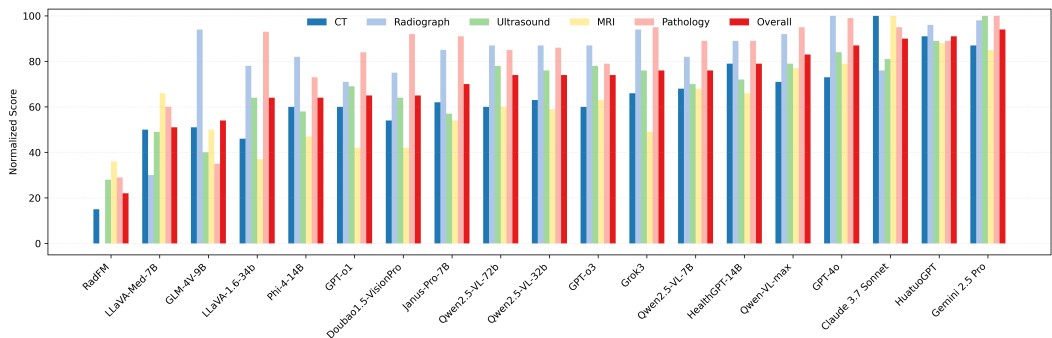

Figure 9: Normalized scores of report generation across five imaging modalities (sorted by overall scores).

Table 17: **BERTScore performance of vision-language models across five imaging modalities.** Best-performing scores are bolded, and second-best scores are underlined. Models follow visual grouping.

| Model | CT | Radiography | Ultrasound | MRI | Pathology | Overall |
|---|---|---|---|---|---|---|
| **Proprietary** | | | | | | |
| GPT-4o | 73/89.42 | **100/92.09** | 84/89.98 | 79/89.11 | 99/90.20 | 87/90.16 |
| GPT-o1 | 60/88.96 | 71/90.79 | 69/89.39 | 42/87.90 | 84/89.69 | 65/89.35 |
| GPT-o3 | 60/88.97 | 87/91.52 | 78/89.73 | 63/88.60 | 79/89.52 | 74/89.67 |
| Claude 3.7 Sonnet | **100/90.35** | 76/91.02 | 81/89.85 | **100/89.80** | 95/90.08 | 90/90.22 |
| Gemini 2.5 Pro | 87/89.89 | 98/91.98 | **100/90.61** | 85/89.30 | **100/90.25** | **94/90.41** |
| Grok3 | 66/88.90 | 94/91.07 | 76/89.35 | 49/87.52 | 95/89.99 | 76/89.77 |
| Doubao1.5-VisionPro | 54/87.67 | 75/89.76 | 64/88.91 | 42/87.45 | 92/89.73 | 65/88.70 |
| Qwen-VL-max | 71/89.71 | 92/91.01 | 79/89.62 | 77/89.38 | 95/89.84 | 83/89.91 |
| **Open-source** | | | | | | |
| Qwen2.5-VL-72b | 60/87.85 | 87/89.99 | 78/89.53 | 60/87.50 | 85/89.71 | 74/88.92 |
| LLaVA-1.6-34b | 46/85.32 | 78/88.66 | 64/88.34 | 37/86.76 | 93/89.81 | 64/87.78 |
| Qwen2.5-VL-32b | 63/88.26 | 87/89.99 | 76/89.31 | 59/87.43 | 86/89.72 | 74/88.94 |
| Phi-4-14B | 60/87.84 | 82/89.12 | 58/87.46 | 47/87.04 | 73/88.26 | 64/87.74 |
| GLM-4V-9B | 51/86.37 | 94/91.05 | 40/86.36 | 50/87.23 | 35/85.48 | 54/87.30 |
| Qwen2.5-VL-7B | 68/88.98 | 82/89.12 | 70/88.91 | 68/88.61 | 89/89.88 | 76/89.50 |
| Janus-Pro-7B | 62/88.15 | 85/89.79 | 58/87.10 | 54/87.28 | 91/89.89 | 70/88.84 |
| **Medical-specific** | | | | | | |
| HuatuoGPT | 91/91.40 | 96/91.65 | 89/89.06 | 88/88.38 | 89/89.37 | 91/90.94 |
| HealthGPT-14B | 79/90.08 | 89/88.79 | 72/88.26 | 66/88.09 | 89/89.08 | 79/88.86 |
| RadFM | 15/81.46 | 0/80.00 | 29/82.85 | 36/83.64 | 29/82.87 | 22/82.17 |
| LLaVA-Med-7B | 50/85.04 | 30/83.01 | 49/84.94 | 66/88.09 | 60/86.01 | 51/85.13 |

# C Prompts

In this section, we present the exact prompts used in our experiments. To ensure model compliance, particularly with recent models that may reject clinical queries, all prompts are framed within an educational context.

## C.1 Multiple-Choice Problems

**System prompt:**
You are a helpful assistant participating in an educational visual reasoning task. Always follow the instructions strictly.

**User prompt:**
**Question:** {}

**Options:**
A) {}
B) {}
C) {}
D) {}
**Instructions:**
Choose the SINGLE best answer by replying with one capital letter (A–D). Do not explain. Do not add extra text.

## C.2 Independent QA

You are participating in an educational exercise based on visual information.

Below is an image typically used in medical training, followed by a question.

**Question:** {}

**Options:**
A) {}
B) {}
C) {}
D) {}

**Instructions:** Based only on the visible structures and general anatomical knowledge, choose the most appropriate answer by replying with a single capital letter (A, B, C, or D). Do not explain your reasoning.

## C.3 Joint QA

**System prompt:**
You are participating in a step-by-step medical reasoning diagnosis task based on interpretation of a medical image.

**User prompt:**
Please answer the following four questions sequentially. Each question builds upon the reasoning of the previous one. Carefully analyze the image and select the most appropriate answer at each step. For each question, choose one capital letter (A, B, C, or D). Do not skip any step.
**1. What imaging modality is used in this image?**
Options:
A. {}
B. {}
C. {}
D. {}
**2. Which organ appears to be abnormal in this image?**
Options:
A. {}
B. {}

C. {}
D. {}

**3. Based on the abnormal organ, what lesion or finding is most clearly visible?**
Options:
A. {}
B. {}
C. {}
D. {}
E. {}
F. {}
G. {}
H. {}

**4. Considering all the above findings, what is the most likely diagnosis?**
Options:
A. {}
B. {}
C. {}
D. {}
E. {}
F. {}
G. {}
H. {}

**Instructions:**
Please reply with your four selected letters in order, separated by commas (e.g., A,C,B,A). Do not provide explanations.

## C.4 Multi-turn QA

**System prompt:**
You are an expert medical AI. You will answer several step-by-step questions about the same medical image. Respond with one capital letter (A–H).

**User (Round 1):**
*(with image)*
1. What imaging modality is used in this image?
Options: A) CT    B) MRI    C) Ultrasound    D) Radiography
**Assistant:**
A
**User:**
`[Same image QA record]` 1. What imaging modality is used in this image?
**Assistant:**
CT

**User (Round 2):**
2. Which organ is shown in this medical image?
Options: A) Liver    B) Pancreas    C) Spleen    D) Kidney
**Assistant:**
A
**User:**
`[Same image QA record]` 2. Which organ is shown in this medical image?
**Assistant:**
Liver

**User (Round 3):**
3. What lesion is visible in this image?
Options: A) Target sign    B) Hepatic steatosis    C) Pancreatic pseudocyst    D) Splenic infarct

**Assistant:**
A

**User (Round 4):**
4. What is the most likely diagnosis?
Options: A) Crohn's disease    B) Acute pancreatitis    C) Splenic rupture    D)
Diverticulitis

## C.5   Report Generation

**System prompt:**
Generate a clinical report based on the image. This is used solely for educational
purposes.

**User prompt:**
Generate a clinical report based on the image. Limit your output to no more than
500 words. *(with image)*
{question}

## C.6   Key Feature Extraction in Generated Reports

Given the following description of a medical image, extract only clinically relevant
information that can be visually determined from the image. This includes both
**normal findings** (e.g., "no lung opacity", "normal heart size") and **abnormal
findings** (e.g., "fracture", "tumor mass"). Exclude any details that cannot be
inferred from the image itself (e.g., patient history, lab values).

*Input:* {text}

Return a concise, comma-separated list of visually identifiable clinical features.
Do not include any irrelevant words or phrases, do not include explanations.

# D Examples for Model Response

In this section, we provide examples of model responses. For multiple-choice questions, models are required to respond with a single capital letter (A, B, C, or D) corresponding to their selected answer. If the response does not follow this format, we perform automatic answer extraction using DeepSeek-V3 [43]. If no valid answer can be extracted, the response is considered *incorrect*.

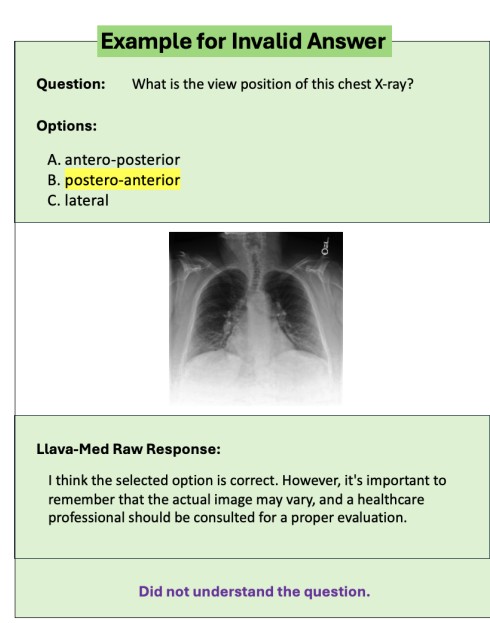

Figure 10: An example of an invalid answer, which is marked as *wrong*. The option highlighted in yellow is the correct answer.

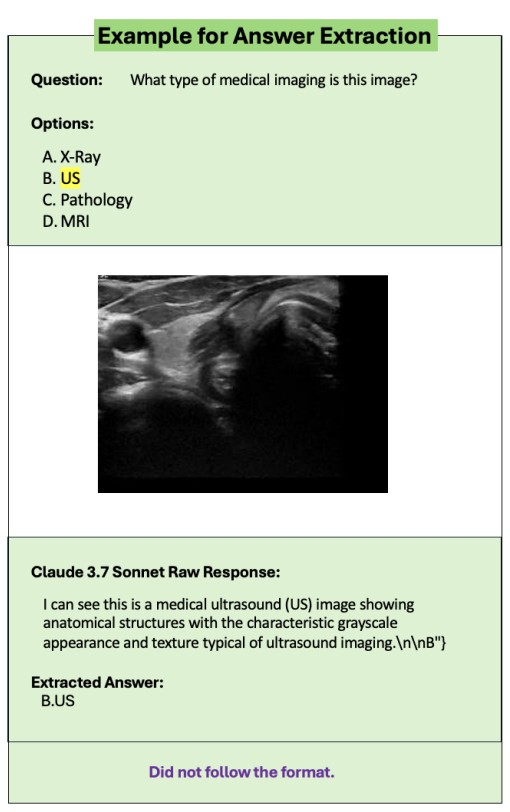

Figure 11: An example of a valid but unformatted answer, which is still marked as *correct*. The option highlighted in yellow indicates the correct answer.

We provide an example of Joint QA below. The formats of Independent QA and Multi-turn QA are similar and thus omitted. Both Independent and Multi-turn QA present questions sequentially; however, in Independent QA, each answer is given without memory of prior interactions, whereas in Multi-turn QA, the model retains memory of previous questions and answers, simulating a step-by-step reasoning process.

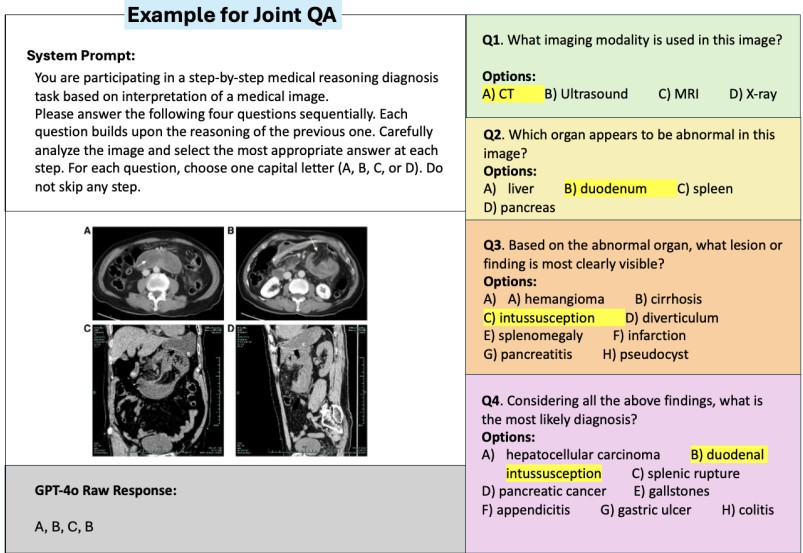

Figure 12: An example of Joint QA, in which the model answers four questions from different reasoning levels simultaneously. The option highlighted in yellow indicates the correct answer.

# E    Examples of Tasks

This section provides detailed examples of our tasks. Choices highlighted in yellow represent the ground truth. In the responses, correct answers are highlighted in green, while incorrect ones are highlighted in red.

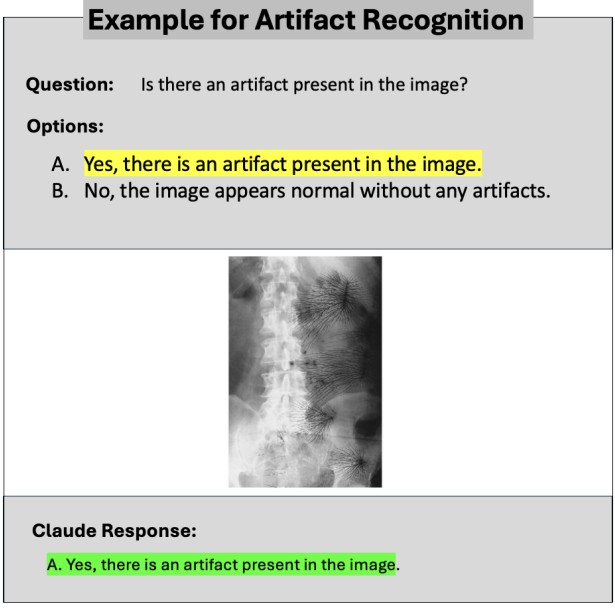

Figure 13: Example for Artifact Recognition

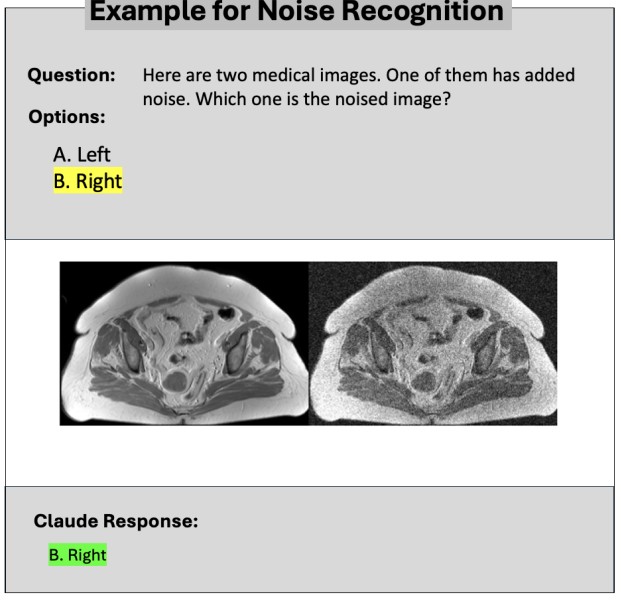

Figure 14: Example for Noise Recognition

**Example for Modality Recognition**

**Question:** What type of medical imaging is this image?

**Options:**
   A.  Radiography
   B.  Ultrasound
   C.  Magnetic Resonance Imaging
   D.  Pathology

**Claude Response:**

A. Radiography

Figure 15: Example for Modality Recognition

**Example for View Recognition**

**Question:** What is the view position of this CT image?

**Options:**
   A.  sagittal
   B.  axial
   C.  coronal

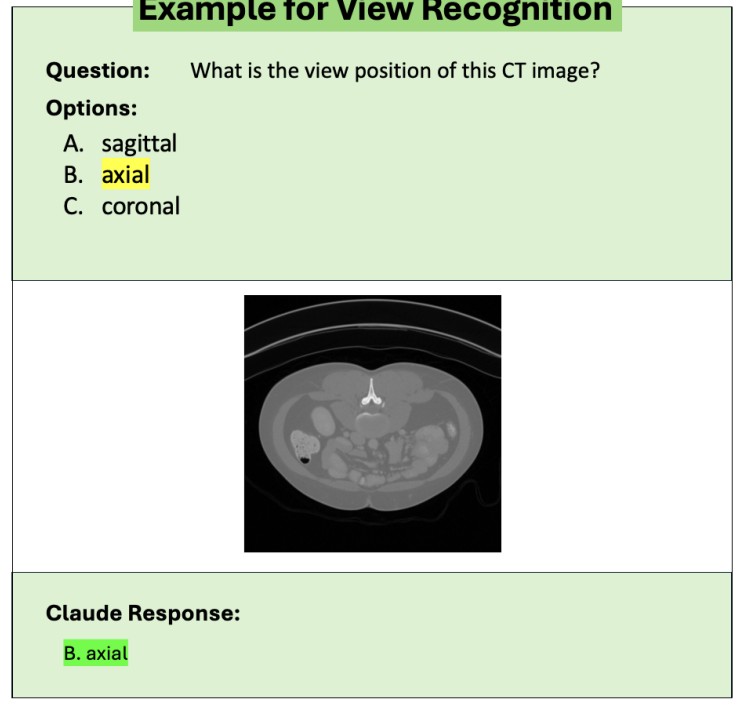

**Claude Response:**

B. axial

Figure 16: Example for View Recognition

**Example for Body Part Recognition**

**Question:** What body part is shown in this MRI?

**Options:**
- A. Head
- B. Chest
- C. Abdomen
- D. Pelvis

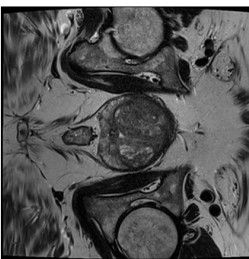

**Claude Response:**

D. Pelvis

Figure 17: Example for Body Part Recognition

**Example for Magnification Recognition**

**Question:** Based on the field of view and visible detail, is this image captured under low-power (<=10x) or high-power (>=20x) magnification?

**Options:**
- A. Low-power field (e.g., overview of tissue, <=10x)
- B. High-power field (e.g., individual cells and nuclei, >=20x)

**Claude Response:**

B. High-power field (e.g., individual cells and nuclei, >=20x)

Figure 18: Example for Magnification Recognition

## Example for Stain Recognition

**Question:**     What is the type of staining used in this image?

**Options:**

  A.  H&E
  B.  Masson's Trichrome
  C.  PAS
  D.  Wright's stain

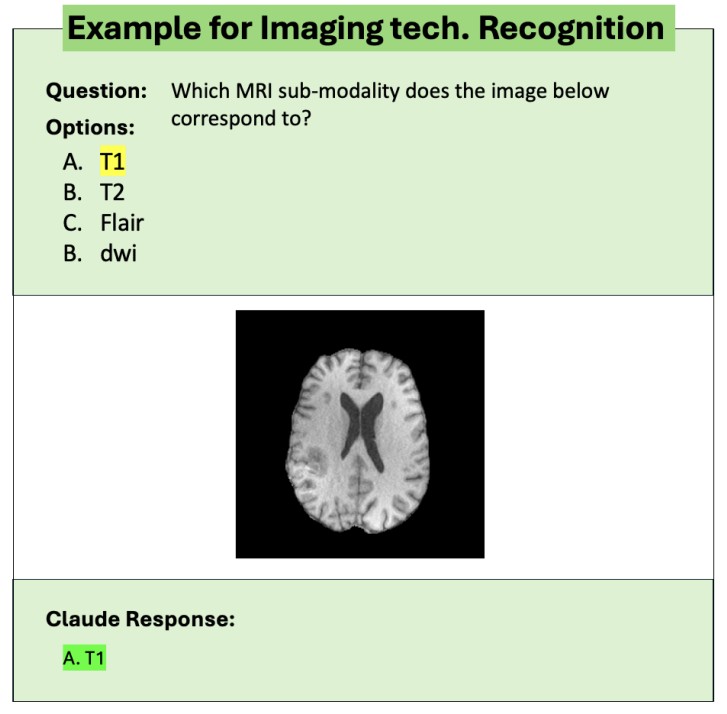

**Claude Response:**

A. H&E

Figure 19: Example for Stain Recognition

## Example for Imaging tech. Recognition

**Question:**     Which MRI sub-modality does the image below correspond to?

**Options:**

  A.  T1
  B.  T2
  C.  Flair
  B.  dwi

**Claude Response:**

A. T1

Figure 20: Example for Imaging Technique Recognition

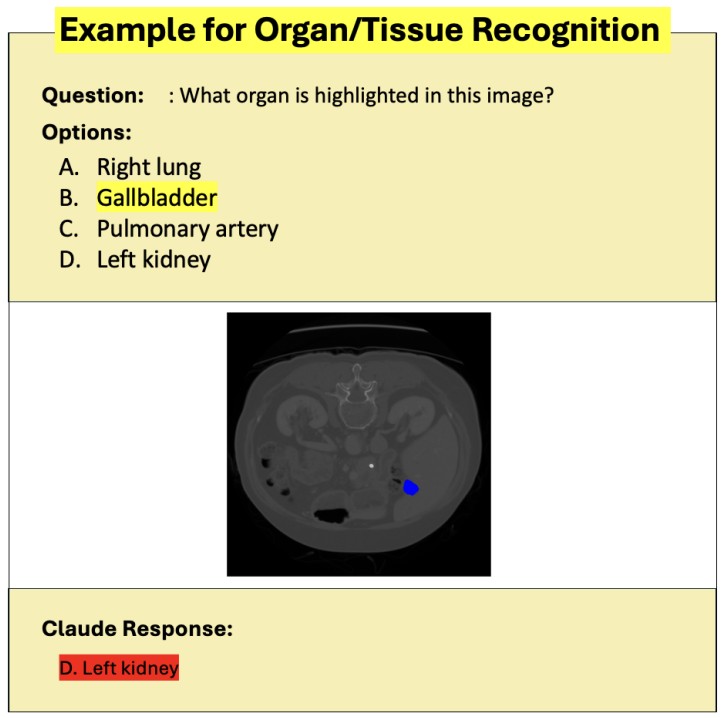

Figure 21: Example for Organ Recognition

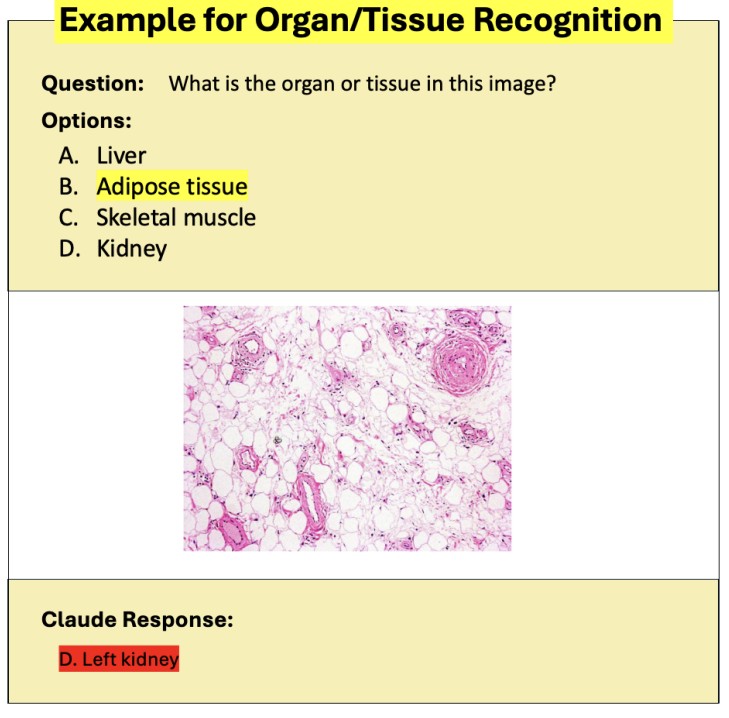

Figure 22: Example for Tissue Recognition

## Example for Organ Location

**Question:**   Which color represents aorta in this image?

**Options:**

  A.  Red
  B.  Green
  C.  Blue
  D.  Yellow

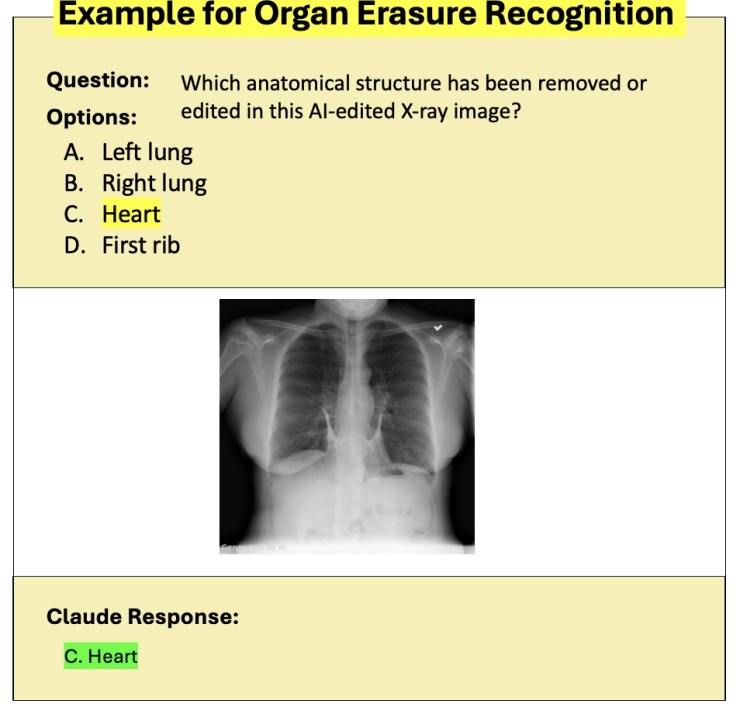

**Claude Response:**

A. Red

Figure 23: Example for Organ Location

## Example for Organ Erasure Recognition

**Question:**   Which anatomical structure has been removed or edited in this AI-edited X-ray image?

**Options:**

  A.  Left lung
  B.  Right lung
  C.  Heart
  D.  First rib

**Claude Response:**

C. Heart

Figure 24: Example for Organ Erasure Recognition

## Example for Lesion Recognition

**Question:** What abnormality is inside the red box in this image?

**Options:**

A. Atelectasis
B. Pneumothorax
C. Cardiomegaly
D. Fibrosis

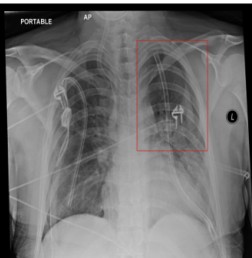

**Claude Response:**

B. Pneumothorax

Figure 25: Example for Lesion Recognition

## Example for Lesion Erasure Recognition

**Question:** This is a pair of CT images, with the original scan on the left and the AI-edited image on the right. Can you identify which type of lesion has been removed or edited in the right image?

**Options:**

A. Lung nodule
B. Lung cancer
C. Pleural effusion
D. Lung infections

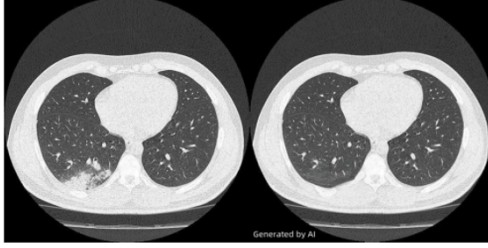

**Claude Response:**

C. Pleural effusion

Figure 26: Example for Lesion Erasure Recognition

## Example for Morphology Description

**Question:** Which of the following best describes the cellular morphology observed in the slide?

**Options:**

A. Spindle cells arranged in tightly packed, parallel fascicles
B. Spindle cells aligned in a whorled, storiform pattern
C. Spindle cells loosely distributed in a disorganized matrix
D. Spindle cells forming solid sheets without architectural arrangement

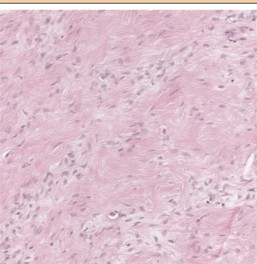

**Claude Response:**

C. Spindle cells loosely distributed in a disorganized matrix

Figure 27: Example for Morphology Description

## Example for Diagnosis

**Question:** What is the most likely histopathological diagnosis for this slide?

**Options:**

A. Interface hepatitis with lymphocytic infiltration
B. Presence of a leukocyte within a hepatocyte
C. Endothelial lifting in a portal venule
D. An eosinophilic leukocyte in the sinusoidal space

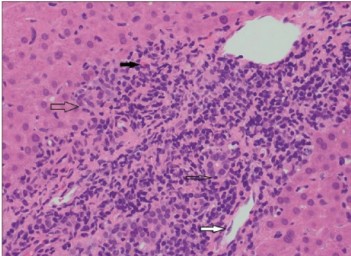

**Claude Response:**

A. Interface hepatitis with lymphocytic infiltration

Figure 28: Example for Diagnosis

