# OpenReview forum: "DrVD-Bench: Do Vision-Language Models Reason Like Human Doctors in Medical Image Diagnosis?"
_NeurIPS.cc/2025/Datasets_and_Benchmarks_Track — NeurIPS 2025 Datasets and Benchmarks Track poster_

### Official Review · Reviewer_iJcf · 2025-06-29

**Rating:** 4
**Confidence:** 4

**Summary:**

The submission introduces DrVD-Bench, a suite of hierarchical benchmarking tasks for medical reasoning in vision-language models. The authors propose a series of benchmarks that evaluate different levels of clinical reasoning, and evaluate a number of existing VLMs on their performance on the newly constructed benchmark.

**Additional Feedback:**

10. For the report generation tasks involving CT and MRI volumes, are the whole volumes available as model context or just singular slices?

**Dataset Code Accessibility:**

Yes

**Dataset Code Comments:**

I was able to load the full dataset using Kagglehub without any issues. The scripts to run the actual benchmarking also seem to be well-documented.

**Ethical Comments:**

The benchmarking suite makes use of existing datasets (see Table 6), so I don't think the submission introduces any significant ethical concerns. The authors also discuss potential limitations on page 9 of their work sufficiently to the best of my knowledge.

**Ethical Considerations:**

No, there are no or only very minor ethics concerns

**Final Justification:**

The authors had a strong rebuttal in my opinion, and were able to address most of my concerns. The inclusion of the pip package installation, together with additional efforts regarding reproducibility and extensibility, make this work of value beyond just the paper itself. The reason why I am choosing to only raise my score 3 --> 4 as opposed to 5 is that the authors are tackling a challenging problem which has a number of inherent limitations as discussed between myself and the authors during the discussion period. The benchmark, while valuable, allows many of these limitations to persist. That being said, I believe I am still in support of this work after the significant improvements made.

**Limitations Weaknesses:**

6. Evaluating whether VLM models can reason like clinicians is an important problem. However, an important component of a clinical benchmarking suite that aims to evaluate this is clinician involvement. Were any clinicians involved in the construction of DrVD-Bench? In particular, I have the following concerns about the dataset from a clinical perspective: (6a) The image quality tasks feel very contrived – for example, in Figure 2, asking which image has added noise is unlikely to be indicative of the medical VL reasoning capabilities of a model. A more meaningful question might be “is this scan interpretable or not” which would require expert labels from clinicians to evaluate this. (6b) Was there any clinician feedback (whether from authors/collaborators or support from prior literature) on the definition of the levels in Figure 2 (to ensure that they are indeed representative of the levels of clinical reasoning)? (6c) is there any clinician evaluation of the overall benchmarking dataset, like providing feedback/annotations on Figure 3 for instance?
7. In addition to the above points, multiple choice QA is generally a poor proxy for clinical reasoning (e.g., see [[1]](https://pmc.ncbi.nlm.nih.gov/articles/PMC10348524/)) and I hope to see the field move away from this benchmarking strategy (e.g. [[2]](https://arxiv.org/abs/2506.11474) and [[3]](https://medarena.ai/login) and Step 3 CCS Cases where information is revealed stepwise and therefore more representative of true reasoning). That being said, I recognize that it MCQA has been a common benchmark in a lot of medical datasets in the past and don't believe this to be a major limitation of this paper especially given that the other benchmarking tasks are generally ok.
8. My understanding is that the source datasets in Table 6 are generally open-source and published from large institutions and therefore introduces potential bias, as minority patients, patients from smaller hospital institutions, and patients from resource-limited countries are likely systematically excluded from the dataset. If this is indeed correct, the manuscript should consider potential augmentation strategies to correct for this, or at least discuss this potential limitation in the conclusion.
9. I feel like an easy explanation for the general trend in Figure 4 is that the data available in DrVD-Bench – many of which are derived from publicly available sources – have already been used to train more recent models. I understand that identifying or correcting for this is not a central component of D&B papers, although I think it is worth including in the Limitations section at the very least.

**Strengths Contributions:**

1. The organ and leisure erasure tasks are novel to the best of my knowledge.
2. Medical reasoning is an important problem, and I'm glad that the problem is being tackled by the authors.
3. The benchmarking experiments included in the manuscript are substantial and well-described. Figure 4 is particularly appreciated and I applaud the authors for their extensive experimental work.
4. As a corollary to point (3), the experiments allow the authors to offer interesting points of discussion (i.e., essentially all of Section 5) that I believe would be of interest to the broader community. As a clinician, I found Section 5.1.2 particular interesting.
5. I appreciate the authors' efforts to stratify the different "levels" of reasoning as documented in Figure 2 and Section 3.2.2.

---

> ### Author Rebuttal · Authors · 2025-07-31
>
> Thank you for the detailed feedback. We are grateful for your recognition of the significance of DrVD-Bench in advancing the evaluation of clinical reasoning in VLMs, and for highlighting the novelty of the organ and lesion erasure tasks. We especially appreciate your positive comments on the thoroughness of our experiments, the value of Figure 4, and the insights provided in Section 5, particularly Section 5.1.2, from a clinical perspective. Your recognition of our efforts to stratify reasoning levels (Figure 2, Section 3.2.2) is also deeply appreciated and encouraging for our continued work.
>
> At the same time, we thank you for your constructive suggestions regarding clinician involvement, the design of image quality tasks, the limitations of multiple-choice QA, dataset bias, and the potential overlap between benchmark and pretraining data. These points are extremely valuable and have prompted us to revise the manuscript with greater clarity, transparency, and clinical relevance. We address each of your comments point-by-point below.
>
> **Response to Limitations Weaknesses 6**
>
> Thank you for raising these important clinical concerns.
>
> **(6a)** We chose objective, verifiable image-quality tasks (such as noise-level detection and artifact identification) because they have a clear ground truth, enabling consistent scoring and benchmarking. In contrast, asking whether an image is “diagnostically usable” would require subjective judgments and consensus labels from multiple experts. Given these practical constraints on data construction and objective scoring, we adopted tasks with unambiguous answers in the current version.
>
> We fully agree that such tasks may better reflect real-world reasoning. We plan to invite specialist radiologists to reach consensus labels and introduce a “diagnostic interpretability” sub-task to better match clinical reality.
>
>  **(6b)** The five-level framework was finalized after extensive discussions with three professors from Tsinghua University School of Medicine—including experts in pathology and oncology—and three MD candidates (including one of our co-authors). They confirmed that the five-step sequence closely mirrors routine clinical workflows. Based on discussion feedback, we added tasks such as organ-erasure identification and morphology description to cover common diagnostic scenarios.
>
> Our structure also aligns with prior literature, such as Waite et al.’s three-stage model [1] which decomposed radiological interpretation into three stages—detection, description and classification/diagnosis, and a newly released benchmark GMAI-MMBench [2] which employs a multi-scale hierarchy to evaluate medical VLMs, echoing our progressive assessment from basic information to final diagnosis.
>
> **(6c)** After construction, we performed a dedicated clinical quality-assurance review of the entire benchmark: under the guidance of our MD-candidate co-author, two attending clinicians—one from Peking Union Medical College Hospital and one from Beijing Tsinghua Changgung Hospital—together with two additional MD candidates from Tsinghua University School of Medicine independently examined every image and its corresponding QA pair.
>
> The review focused on: (i) Image usability: confirming that each image meets diagnostic-quality standards or, in the “image-quality” tasks, that the synthetic artifact/noise level matches the label; and (ii) Content accuracy: verifying the clinical correctness of every question and ground-truth answer.
>
> [1] Waite et al. Analysis of Perceptual Expertise in Radiology—Current Knowledge and a New Perspective. Front Hum Neurosci (2019).
>
> [2] OpenGVLab. GMAI-MMBench: General Medical Artificial Intelligence Multimodal Benchmark. NeurIPS (2024).
>
> **Response to Limitations Weaknesses 7**
>
> Thank you for your thoughtful comment and references. We agree that multiple-choice QA is not a perfect proxy for clinical reasoning. That said, it offers practical benefits, such as objective answers, consistent scoring, and lower annotation cost, which made it a feasible choice for the current version of DrVD-Bench.
>
> To move beyond its limitations, we also included free-text report generation as a complementary task in this study (Figure 3). While current evaluation relies on BERTScore for simplicity and comparability, we are actively developing more robust metrics that assess factual accuracy, image-grounding, and reasoning coherence. We see this work as a starting point, and your suggestions strongly support our plan to expand toward richer, more clinically faithful reasoning benchmarks in future iterations.
>
> **Response to Limitations Weaknesses 8**
>
> Thank you for raising this important point. You are right that most datasets in Table 6 come from large academic centers in Europe, North America, and Asia, which may under-represent minority groups, smaller hospitals, and resource-limited regions. We will explicitly acknowledge this limitation in the revised manuscript, as it may lead to overestimating model generalizability in real-world, underserved settings. We also plan to expand DrVD-Bench with additional data from rare diseases and under-represented ethnic groups, prioritizing images from multi-ethnic cohorts, lower-tier equipment, and small-scale institutions.
>
> **Response to Limitations Weaknesses 9**
>
> Thank you for pointing out the potential “training–test overlap” issue. Since DrVD-Bench relies on publicly available datasets, it’s possible that some recent models have seen parts of this data during pretraining, which may partially account for the performance trends in Figure 4. We will include this as a clear limitation in the revised manuscript.
>
> To help readers better assess this risk, we will add a table in the appendix listing the public release dates and DrVD-Bench proportions of each source dataset, along with available training details for the evaluated models. This will make any potential data leakage more transparent and help contextualize the models’ generalization results.
>
> **Response to Additional Feedback 10**
>
> Thank you for the question. For the CT and MRI report generation tasks, DrVD-Bench currently provides models with a single or a few representative slices rather than full 3D volumes. This design reflects two main considerations. First, most generalist medical VLMs today don’t yet support direct processing of full 3D data. Second, our data sources (e.g., PubMedVision) typically include only key slices published alongside diagnostic text, not full scans. This strategy also reflects certain real-world clinical workflows, where radiologists often make preliminary assessments or discuss findings in academic meetings based on representative slices rather than reviewing every frame in the entire series.
>
> We acknowledge that this simplification cannot fully substitute for reasoning over complete 3D volumes. In future iterations of DrVD-Bench, as more high-quality, expert-annotated 3D imaging datasets become publicly available and general-purpose MedVLMs gain the ability to process full volumetric inputs, we plan to incorporate volume-level report generation tasks to more comprehensively evaluate model reasoning performance on 3D data.

---

> > ### Comment · Reviewer_iJcf · 2025-08-03
> >
> > I thank the authors for their detailed rebuttal, and have reviewed their comments in addition to the comments from the other reviewers. I appreciate that the authors are willing to include additional discussion regarding the limitations raised in my initial review. Highlighting comments 6b-c made by the authors regarding the clinical verification of this dataset will also further strengthen this work.
> >
> > More broadly, it seems like many of these limitations are unlikely/impossible to address in a single round of revisions. I believe this is more a reflection of the challenge of constructing the "perfect" benchmark for medical reasoning, and second Reviewer 36gY's comment that the authors performed a great job and commendable effort putting together a benchmark that may not be "perfect," but is still promising and may be of value to the greater community given additional time.
> >
> > The ability for the authors (and hopefully future researchers!) to extend upon this work and address some of the limitations (i.e., points 7, 8, 9, and 10 in my initial review) in a "living" version of this benchmark is dependent on how extensible the authors' dataset, code, and overall implementation is. I believe that the quality of the dataset and code released is consistent with that of other recent benchmarks (e.g., [Open Medical-LLM Leaderboard](https://github.com/hendrycks/test/tree/master) and [HealthBench](https://github.com/openai/simple-evals/tree/main)). That being said, it may be worth considering how to improve the long-term utility + reproducibility of this benchmark, i.e. dockerizing as in [MedAgentBench](https://github.com/stanfordmlgroup/MedAgentBench) and documentation + pip package release as in [MedHELM](https://crfm.stanford.edu/helm/medhelm/latest/), which I feel are strong positive examples for benchmark implementations to follow.
> >
> > If the authors are able to implement some/all of these code changes to improve reproducibility, together with the additional discussion on limitations in the manuscript text as already addressed in the rebuttal, I would be in greater favor of accepting this work. However, I also recognize that asking for such major code changes is by no means expected in such a short discussion period. Given these reasons, I maintain my score for now, although will make sure to re-evaluate the code base prior to submitting the Final Justification and make any updates to my score as appropriate.

---

> > > ### Author Response · Authors · 2025-08-04
> > >
> > > Thank you again for taking the time to review our work and for providing such constructive feedback. In response to your valuable new suggestions on code reproducibility, we have already begun making improvements and plan to complete the first round within the current discussion period.
> > >
> > > We have launched three initiatives:
> > >
> > > - Modularization and pip release: We are refactoring core components—question generation, model inference, answer mapping, and metric computation—into separate modules and packaging them as a one-click, pip-installable library to facilitate easy reproduction and extension.
> > > - Interactive tutorials: A comprehensive Jupyter notebook covering data download, model inference, and metric evaluation will be provided to give new users a minimal, reproducible example.
> > > - Third-party validation: External researchers (independent of the author team) are running the full pipeline in fresh environments; their feedback is helping us fix edge cases and improve robustness. Anonymous testing reports and change logs will be shared before the camera-ready deadline.
> > >
> > > Because NeurIPS policy prevents updating the public GitHub repository during review, we would like to share a pip install command and usage guide for the package with you before the discussion phase ends. All changes will be officially released as a versioned update after the review, accompanied by full documentation.
> > >
> > > Finally, as you highlighted in points 7–10, our current work (and similar efforts) still falls short in some limitations. These are long-term challenges we are committed to addressing, and we envision DrVD-Bench as a “living” resource that evolves with community needs. The revised manuscript will include a dedicated section that systematically discusses these limitations and outlines a clear roadmap for future iterations.
> > >
> > > Thank you once again for your thoughtful and constructive feedback. We truly value your input and We look forward to any further feedback you may have as we continue refining DrVD-Bench.

---

> > > ### Author Response · Authors · 2025-08-08
> > > **Release of DrVD-Bench on Python Package Index (PyPI)**
> > >
> > > Dear Reviewer,
> > >
> > > As part of our ongoing effort to address your valuable feedback regarding reproducibility and long-term utility, we’re pleased to share that the DrVD-Bench toolkit is now available on the Python Package Index (PyPI). This completes the first round of improvements we committed to during the discussion period.
> > > This update aims to make the benchmark easier to install, understand, and extend. The following enhancements have been implemented:
> > >
> > > - **Modularization & PyPI release**: The toolkit’s core components have been refactored into modular Python packages and released as drvd-bench on PyPI, enabling one-click installation and smooth extension.Due to NeurIPS anonymity policy, we are not permitted to post a direct link here.
> > >
> > > - **Interactive tutorial**: A full Jupyter notebook demonstrating data download, model inference, and metric evaluation is bundled with the PyPI page. In compliance with NeurIPS review policy, the public GitHub mirror will be made available immediately after the review period ends.
> > >
> > > - **Third-party validation**: Independent researchers have successfully run the toolkit in fresh environments. Their feedback has helped us resolve edge cases and improve robustness. Anonymized test reports and change logs will be included in the camera-ready version.
> > >
> > > You can now install the package via: **pip install drvd-bench**. Below is a minimal usage guide for quick verification.
> > > We sincerely hope these improvements reflect our commitment to building an extensible, community-driven benchmark.
> > >
> > > Thank you again for your thoughtful and constructive feedback!
> > >
> > > ##minimal usage guide for quick verification##
> > > ### 0. Installation
> > >
> > > ```bash
> > > pip install drvd-bench
> > > ```
> > >
> > > ---
> > >
> > > ### 1. Function `get_drvd_data`
> > >
> > > #### Functionality
> > > Reads JSONL, generates the final prompt, and returns an iterator of (img_path, prompt, origin_data).
> > >
> > > #### Note
> > > This code only returns an iterator for retrieving data and does not include model inference.
> > >
> > > #### Parameters
> > > - `jsonl_path (str | Path)`: Path to the dataset JSONL file
> > > - `image_root (str | Path)`: Root directory of the dataset
> > > - `data_type ({“single”, “joint”})`: Question type, default “single”. Use “joint” only for `joint_qa.jsonl`; all other JSONL files should use “single”
> > > - `verbose (bool)`: Whether to show a progress bar; default is True
> > >
> > > #### Returns
> > > Iterator[(img_path:str, prompt:str, record:dict)]
> > >
> > > #### Example
> > > ```python
> > > from drvd_bench import get_drvd_data
> > > for img, prompt, original_data in get_drvd_data(
> > >     "test_drvd_pypi/data/visual_evidence_qa.jsonl",
> > >     "test_drvd_pypi/data",
> > >     data_type="single"
> > > ):
> > >     print(img, prompt[:80] + "…")
> > > ```
> > >
> > > ---
> > >
> > > ### 2. Function `map_result`
> > >
> > > #### Functionality
> > > Calls the DeepSeek Chat API to map open-ended responses to options and writes back to JSONL.
> > >
> > > #### Note
> > > Use only when the model instruction-following capability is insufficient and for non-report generation tasks. Not needed if the model instruction-following is strong.
> > >
> > > #### Parameters
> > > - `api_key (str)`: DeepSeek API Key; refer to the DeepSeek official website for details
> > > - `input_path (str | Path)`: Path to the original prediction JSONL file
> > > - `output_path (str | Path)`: Path to save the mapped JSONL file
> > > - `base_url (str)`: Private deployment URL; typically does not need modification
> > > - `show_preview (int)`: Preview the first N results; default is 5
> > >
> > > #### Example
> > > ```python
> > > from drvd_bench import map_result
> > > map_result(
> > >     api_key="sk-…",
> > >     input_path="pred.jsonl",
> > >     output_path="mapped.jsonl"
> > > )
> > > ```
> > >
> > > ---
> > >
> > > ### 3. Function `compute_choice_metric`
> > >
> > > #### Functionality
> > > Calculates accuracy for multiple-choice questions.
> > >
> > > #### Note
> > > Your JSONL file must contain an `answer` key for the original data’s answer. The model’s result must be stored in the key `model_response` and should be a single option, such as A, B, C, D, etc.
> > >
> > > #### Parameters
> > > - `jsonl_path (str | Path)`: Path to the prediction result JSONL file
> > > - `mode ({“single”, “joint”})`: Question type, default “single”. Use “joint” only for `joint_qa.jsonl`; all other multiple-choice JSONL result files should use “single”
> > >
> > > #### Example
> > > ```python
> > > from drvd_bench import compute_choice_metric
> > > compute_choice_metric(
> > >     "visual_evidence_qa_pred.jsonl",
> > >     mode="single"
> > > )
> > > ```
> > >
> > > ---
> > >
> > > ### 4. Function `compute_report_generation_metric`
> > >
> > > #### Functionality
> > > Computes BLEU and BERTScore F1 for report generation tasks.
> > >
> > > #### Note
> > > Your JSONL file must contain an `answer` key for the original data’s answer. The model’s result must be stored in the key `model_response`.
> > >
> > > #### Parameters
> > > - `api_key (str)`: DeepSeek API Key
> > > - `json_path (str | Path)`: Path to the JSONL result file containing `answer` / `model_response` fields for report generation
> > > - `base_url (str)`: Private deployment URL; typically does not need modification
> > >
> > > #### Example
> > > ```python
> > > from drvd_bench import compute_report_generation_metric
> > > compute_report_generation_metric(
> > >     api_key="sk-…",
> > >     json_path="report_generation_preds.jsonl"
> > > )
> > > ```

---

> > > > ### Comment · Reviewer_iJcf · 2025-08-09
> > > >
> > > > I thank the authors for their hard work on this - I was able to take a closer look at the updated code available and greatly appreciate the efforts made. In particular, I feel more confident now that this proposed benchmark will become a benchmark that the research community can build on in the future, and even work towards addressing the (inherently challenging!) limitations discussed in earlier comments. I am updating my score from 3 --> 4 to indicate my support of this work.

---

### Official Review · Reviewer_36gY · 2025-07-02

**Ethics Flags:** Improper research involving human sub…
**Rating:** 6
**Confidence:** 4

**Summary:**

This paper presents DrVD-Bench, a novel hierarchical benchmark to evaluate the clinical reasoning process of VLM. It features a hierarchical framework assessing reasoning from visual evidence comprehension to diagnosis. It also supports a report generation evaluation. Its modules are designed to assess models' stepwise reasoning versus reliance on shortcuts, and include erasure tasks to compel models to reason from visible evidence.

**Additional Feedback:**

I will decrease my rate if my concerns can not be solved, I love this work, so i need this work to be perfect.

**Dataset Code Accessibility:**

Yes

**Ethical Considerations:**

No, there are no or only very minor ethics concerns

**Final Justification:**

6 score.

**Limitations Weaknesses:**

However, i found some hard negative problem regarding to its evaluation metrics, where as a benchmark paper this is very important.

1. The evaluation metrics for reports are too simplistic and lack assessment of hallucinations, which may lead to the frequent use of visual evidence comprehension tasks as benchmarks while report generation evaluation is overlooked. My questions are: 1. Referring to FactScore, if factual benchmarks are added to the reports themselves, could this reflect more effective information? 2. The definition of Normalized BERTScore is not suitable for other researchers to use and compare.

2. The analysis of the causes of multi-turn reasoning failures is insufficient, resulting in unsurprising results. For instance, it is not clear whether it is due to the accumulation of errors mentioned in the text, the catastrophic forgetting of <image> token caused by multi-turn, the lack of reference to image in the prompt template, or the introduction of additional priors by the higher-level Question in Joint QA. Analysis cannot be drawn without experiments.

**Strengths Contributions:**

**I love this work actually, because of its substantially hard work.**

1. The work is complete, and the motivation is directed towards practical problems. The 3-modules and 5-level design provides a stepwise evaluation, which has a positive effect on trustworthy medical VLM.

2. Visual Evidence Comprehension provides reliable metrics.

3. The analysis perspectives are diverse, and the experiments are thorough. Especially, the part of "overdiagnosis without understanding" presents key insights.

---

> ### Author Rebuttal · Authors · 2025-07-31
>
> We sincerely thank you for the encouraging comments. We are grateful for your appreciation of the effort behind DrVD-Bench. Your positive feedback on the “overdiagnosis without understanding” analysis and the value of the Visual Evidence Comprehension module reinforces our belief in the importance of evaluating trustworthy and interpretable reasoning in medical VLMs. We also deeply appreciate your critical observations regarding the evaluation metrics for report generation and the causes of multi-turn reasoning failure. These are insightful points that have prompted us to improve the benchmark in both design and analysis. We address each of your concerns point-by-point in the following responses.
>
> **Response to Limitations Weaknesses 1**
>
> Thank you for highlighting this critical point. we have added FactScore to directly assess hallucinations in generated reports.
>
> Our approach first extracts atomic clinical facts from both the original reference reports and model-generated reports using DeepSeek-v3. We then employ GPT-4o to evaluate fact-level matches, computing recall (coverage of true facts) and specificity (avoidance of hallucinated facts) for each model. Due to time constraints and the instability of official APIs, we performed the experiment on four representative models (Gemini2.5pro, Claude 3.7 Sonnet, Qwen 2.5 VL 7B, and HuatuoGPT), and the preliminary results (Table R1) reveal consistent shortcomings:
>
> - Low recall (≤ 0.13) indicates that only a small fraction of clinically relevant facts are correctly mentioned by the models.
> - Low specificity (≤ 0.09) highlights frequent hallucinations, with many facts in generated reports not supported by the image or original report.
>
> These findings validate your concern and confirm that factual consistency remains a major bottleneck for current VLMs in clinical reporting. The full results and implementation will be included in the revision.
>
> Regarding the normalization of BERTScore, we used it for easier cross-model comparison in tables across models in tables. To ensure transparency and usability, we have included raw BERTScore values in the appendix so that future studies can make direct and reproducible comparisons.
>
> **Table R1: FactScore Results: Factual Consistency of Report Generation**
>
> | Model            | Recall (std) | Specificity (std) |
> |------------------|--------------|-------------------|
> | Gemini2.5pro     | 0.13 (0.16)  | 0.07 (0.10)       |
> | Claude 3.7 Sonnet| 0.06 (0.11)  | 0.03 (0.05)       |
> | Qwen 2.5 VL 7B   | 0.08 (0.13)  | 0.05 (0.09)       |
> | HuatuoGPT        | 0.13 (0.16)  | 0.09 (0.17)       |
>
> **Response to Limitations Weaknesses 2**
>
> Thank you for highlighting the need for deeper analysis of multi-turn reasoning failures. To isolate the effect of error accumulation from other factors you mentioned (e.g., [object Object] token issues, missing image references, and priors introduced by higher-level question), we added three controlled conditions:
> - **Baseline**: the model proceeds with its own prior answers throughout the dialogue;
> - **Insert-Correct**: we overwrite all low-level answers with ground-truth labels before higher-level questions;
> - **Insert-Error**: we replace them with random incorrect labels.
>
> Due to time constraints and the instability of official APIs, we here report results on three representative models (Qwen2.5-VL-72B, Claude, and GPT-4o). As shown in the results (Table R2), we found that diagnosis accuracy improves substantially when earlier answers are corrected (e.g., +28 points for Qwen), and drops when earlier answers are wrong (–15 points), confirming that error accumulation is the primary cause of performance degradation. Intermediate tasks like Body Part and Organ are less affected (±6%), indicating that dependencies grow stronger in deeper reasoning stages.
>
> Besides, since all settings receive identical prompts, image references, and input lengths, we rule out catastrophic forgetting or prompt format issues as major factors. These findings suggest that error accumulation is a key driver of multi-turn failures, while other factors such as catastrophic forgetting or prompt formatting have minimal effect under controlled conditions.
>
> **Table R2 – Impact of Low-Level Answer Accuracy on Multi-turn QA Performance (Baseline / Insert-Correct / Insert-Error)**
>
> | Model              | Modality   | Body Part  | Organ      | Lesion      | Diagnosis   |
> |--------------------|:----------:|:----------:|:----------:|:-----------:|:-----------:|
> | Qwen2.5-VL-72B| 99/99/99   | 85/84/84   | 57/58/55   | 47/60/43    | 37/65/22    |
> | Claude 3.7 Sonnet| 99/99/99   | 83/84/77   | 52/52/46   | 36/50/32    | 30/60/19    |
> | GPT-4o| 99/99/99   | 85/86/85   | 55/55/54   | 45/52/44    | 40/67/33    |
>
> In addition, we had anticipated the risk that Joint QA might enable models to infer lower-level answers from higher-level question priors. To avoid this, we took specific precautions during benchmark construction in the original work:
> - At the Organ level, only four common organs are presented in randomized order to reduce predictability.
> - At the Lesion level, each organ maps to two candidate lesions, and these lesions are deliberately cross-shared among different organs to prevent back-inference.
> - At the Diagnosis level, we include eight balanced, shuffled diagnoses that do not uniquely correspond to any specific lesion or organ.
>
> These design choices ensure that models cannot exploit structural cues or option alignment to shortcut the reasoning process. As a result, we believe that the observed performance in Joint and Multi-turn QA settings faithfully reflects the models’ actual step-by-step reasoning ability.

---

> > ### Comment · Reviewer_36gY · 2025-08-05
> > **Good luck👋**
> >
> > Authors have solved my concerns. 5$\rightarrow$6 now.

---

> > > ### Author Response · Authors · 2025-08-08
> > > **Thank you for your positive feedback**
> > >
> > > We truly appreciate your time and support, and we will incorporate the points you raised into the revised manuscript.

---

### Official Review · Reviewer_hVkD · 2025-07-04

**Ethics Flags:** Data privacy, copyright, and consent
**Rating:** 5
**Confidence:** 4

**Summary:**

The authors propose DrVD-Bench, a benchmark to evaluate whether Vision Large Language Models in medical imaging follow stepwise clinical reasoning. It includes three modules: visual evidence comprehension, reasoning trajectory assessment, and report generation. Tasks are structured across five clinical levels, from modality recognition to diagnosis, with organ/lesion erasure used to test reliance on true visual cues.

**Dataset Code Accessibility:**

Yes

**Dataset Code Comments:**

Detailed information is listed in the Dataset Reviewer Report

**Ethical Considerations:**

No, there are no or only very minor ethics concerns

**Final Justification:**

Thank the authors for their efforts. As most of my concerns have been addressed I raised my rating to 5 – Accept.

**Limitations Weaknesses:**

1.	Table 4 and the supplementary tables do not clearly indicate which QA setting (Independent, Joint, or Multi-turn) was used for each task.
2.	While Table 5 reports overall accuracy across different QA types, it is unclear why models such as Gemini and Qwen exhibit large performance gaps between Independent and Multi-turn QA in Modality. For the first question in a sequence, both formats should theoretically be equivalent, as no prior context is available.
3.	In the Multi-turn QA setting, it remains unclear whether answers to earlier low-level questions, whether correct or incorrect, impact the model’s performance on subsequent high-level reasoning. Could you provide statistical analysis to examine such potential dependencies across reasoning levels?
4. The evaluation could be further enriched by including a wider range of medical VLLMs, such as Med-PaLM 2[1] and PathOrchestra[2]. Additionally, incorporating disease-specific VLLM like MOME[3] or DeepDR-LLM[4] may provide deeper insights into how specialized knowledge impacts reasoning performance.

[1]Singhal, K., et al., Toward expert-level medical question answering with large language models. Nature Medicine, 2025
[2]Yan, F., et al., Pathorchestra: A comprehensive foundation model for computational pathology with over 100 diverse clinical-grade tasks. arXiv 2025
[3]Luo, L. et al., A large model for non-invasive and personalized management of breast cancer from multiparametric MRI. Nature Communications, 2025
[4]Li, J., Guan, et al., Integrated image-based deep learning and language models for primary diabetes care. Nature medicine, 30(10), 2886-2896.

**Strengths Contributions:**

1.	DrVD-Bench covers diverse diagnostic stages, from image quality to final diagnosis, allowing fine-grained performance analysis across reasoning levels.
2.	The proposed multi-level task design effectively evaluates the reliability of LLMs’ reasoning processes by assessing their performance across progressively structured clinical steps, from basic recognition to diagnostic decision-making.
3.	Results reveal key limitations in current VLMs for Medical Image Diagnosis, offering guidance for future model development.

---

> ### Author Rebuttal · Authors · 2025-07-31
>
> We sincerely thank the reviewer for the constructive feedback. We greatly appreciate your recognition of DrVD-Bench’s contributions in enabling fine-grained evaluation across diagnostic stages and in revealing limitations in current VLMs for medical imaging. Your positive remarks on the multi-level task design and its alignment with stepwise clinical reasoning are especially encouraging, as this was a central motivation behind our benchmark. We also value your detailed suggestions for improvement, including clarifying QA settings, analyzing Multi-turn dependencies, and expanding model coverage, which have helped us further refine the benchmark and strengthen its clinical relevance. We address each of your comments point by point below.
>
> **Response to Limitations Weaknesses 1**
>
> Thank you for pointing this out. Except for the Reasoning Trajectory Assessment experiment (Table 5), which explicitly compares Independent, Joint, and Multi-turn QA settings, all other results in Table 4 and the supplementary tables are based on the Independent QA format. To eliminate any ambiguity, we will revise the table titles to include “(Independent QA)” and clarify this in the Methods section and table captions, ensuring readers can immediately identify the QA setting for each result.
>
> **Response to Limitations Weaknesses 2**
>
> Thank you for your careful observation. As you correctly noted, the first-round “Modality” question should yield equivalent results in the Independent and Multi-turn QA settings. Upon rechecking our experiment logs, we identified a transcription error: the Multi-turn accuracies for Gemini and Qwen in Table 5 were mistakenly taken from an intermediate column during aggregation. The corrected scores are nearly identical to their Independent counterparts (difference < 0.5 percentage points) and do not affect our conclusions.
>
> We have thoroughly verified all other results and confirmed this was the only reporting error. In the revision, we will update Table 5 and the supplementary tables accordingly, and will also include full per-task results and reproduction scripts in the appendix. We sincerely appreciate your attention to detail, which helped us improve the rigor of our work.
>
> **Response to Limitations Weaknesses 3**
>
> Thank you for suggesting this insightful analysis. To evaluate whether early-stage answers affect downstream reasoning in the Multi-turn QA setting, we added the following controlled experiments. Due to time constraints and the instability of official APIs, we here report results on three representative models (Qwen2.5-VL-72B, Claude, and GPT-4o). Specifically, we compared:
> - **Baseline**: the model proceeds with its own prior answers throughout the dialogue;
> - **Insert-Correct**: we overwrite all low-level answers with ground-truth labels before higher-level questions;
> - **Insert-Error**: we replace them with random incorrect labels.
>
> **Table R1: Impact of Low-Level Answer Accuracy on Multi-turn QA Performance  (Baseline / Insert-Correct / Insert-Error)**
>
> | Model              | Modality   | Body Part  | Organ      | Lesion      | Diagnosis   |
> |--------------------|:----------:|:----------:|:----------:|:-----------:|:-----------:|
> | Qwen2.5-VL-72B | 99/99/99   | 85/84/84   | 57/58/55   | 47/60/43    | 37/65/22    |
> | Claude         | 99/99/99   | 83/84/77   | 52/52/46   | 36/50/32    | 30/60/19    |
> | GPT-4o         | 99/99/99   | 85/86/85   | 55/55/54   | 45/52/44    | 40/67/33    |
>
> The results (Table R1) reveal several key patterns:
> - For the first-round **Modality** question (which has no prior context), all three conditions yield identical accuracy, confirming that context injection does not affect context-free queries.
> - Starting from the **Lesion** level, the accuracy of early answers significantly impacts downstream performance. For example, Qwen-VL-2.5 achieves **+29%** improvement in Diagnosis accuracy under *Insert-Correct*, and **–14%** under *Insert-Error*. Claude and GPT-4o exhibit similar trends.
> - Performance on intermediate tasks such as **Body Part** and **Organ** is only modestly affected (within ±6%), indicating that early answer correctness has limited influence at these levels. Taken together, these results suggest that context dependency becomes increasingly pronounced at deeper stages of clinical reasoning.
>
> These findings confirm that VLMs do exhibit context-dependent behavior in clinical reasoning, and suggest that accuracy in early steps is critical for reliable downstream interpretation. We will include this new analysis in the revision to enrich the discussion of multi-step reasoning dependencies.
>
> **Response to Limitations Weaknesses 4**
>
> Thank you for the thoughtful suggestion to expand the model coverage in our evaluation. We would like to clarify a few points:
>
> - **Med-PaLM 2** is a text-only LLM without visual input capability, and therefore falls outside the scope of our multimodal benchmark.
> - **PathOrchestra** has not yet been publicly released, so external evaluation is currently not feasible.
> - **MOME** uses an encoder-only architecture tailored for binary classification, which is incompatible with our open-ended QA tasks.
> - **DeepDR-LLM** is a disease-specific model trained solely on RGB fundus images for diabetic retinopathy (not present in our benchmark), limiting its generalizability across modalities.
>
> Despite these constraints, we conducted supplementary experiments on the latest releases and identified **Patho-R1 (May 2025)** [1]—a pathology-specific VLM fully compatible with our pipeline. We therefore include Patho-R1 as a substitute modality-specific baseline; its partial results are presented in Table R2 below and will be added to the updated results table in the revised manuscript.
>
> As a pathology-specific VLM, Patho-R1 performs markedly better on pathology images than on CT, X-ray, ultrasound, or MRI (Table R2). Nevertheless, its overall score still trails the top general-purpose models reported in the paper, with especially large gaps in cross-modal anatomical localization and lesion reasoning. This shortfall is likely due to Patho-R1’s relatively small parameter size and a training set heavily skewed toward pathology images, leaving it with limited exposure to other modalities and thus constrained generalization.
>
> We also emphasize that DrVD-Bench is designed to be readily extensible, and we hope it can serve as a common platform for evaluating future medical VLMs as they become publicly available.
>
> **Table R2: Partial Results of Patho-R1 Performance on DrVD-Bench**
>
> | Modality   | Image Quality | Basic Info | Anatomy Level | Lesion Level | Diagnosis |
> |------------|---------------|-----------|---------------|--------------|-----------|
> | CT         | 0.50          | 0.60      | 0.32          | 0.35         | 0.60      |
> | X-ray      | 0.69          | 0.54      | 0.34          | 0.25         | 0.25      |
> | Ultrasound | 0.50          | 0.51      | 0.21          | 0.32         | 0.35      |
> | MRI        | 0.55          | 0.45      | 0.31          | 0.42         | 0.55      |
> | Pathology  |       \        | 0.79      | 0.40          | 0.54         | 0.46      |
> | **Overall**| **0.56**      | **0.58**  | **0.32**      | **0.38**     | **0.44**  |
>
> [1] Zhang, W., *et al.* (2025, May). *Patho-R1: A Multimodal Reinforcement Learning-Based Pathology Expert Reasoner.* Preprint at arXiv (arXiv:2505.11404).

---

### Official Review · Reviewer_J5rP · 2025-07-20

**Rating:** 4
**Confidence:** 5

**Summary:**

The authors compile a number of public medical datasets. They extract a selected number of samples from these data (it is not clear how many), and then select 2D slices from 3D data (e.g. MRI, CT and ultrasound video) or sub-areas from large 2D scans (e.g. pathology). They also integrate images from datasets that have scraped figures from medical publications. This results in a set of approximately 5,500 images. For these images, they generate questions and answers. The questions follow a hierarchical order, e.g., "Are there any artefacts present?", "What is the MRI sequence type?", "What organ is at the centre of the image?", "Describe any lesions present" and finally "Structure all of the above in a report". There are around 8,000 of these question-answer pairs for the 5,500 images, covering 20 tasks (machine learning tasks?), 20 diagnostic categories (diagnostic tasks?) and five modalities.  They use this dataset to compare the performance of around 20 LLMs, observing human-like reasoning (medical doctors'?) in some of the tasks and datasets.

**Additional Feedback:**

See above. In general I would encourage the authors to take 3D image data as input to downstream VLMs, as selected 2D images will never suffice in almost all of the diagnostic tasks the 3D data were acquired for.

**Dataset Code Accessibility:**

No

**Dataset Code Comments:**

Code for the metrics used are not available.

Access to the dataset is restricted (in Github, Kaggle) at this point. It is not clear whether the selected images could be backtraced to the original public medical datasets (important for choosing fresh training or test sets, for example).

**Ethical Comments:**

No comments.

**Ethical Considerations:**

No, there are no or only very minor ethics concerns

**Final Justification:**

See my final comment to the authors:

>     DrVD-Bench is intended as a benchmark to evaluate the diagnostic reasoning ability of pretrained VLMs, rather than a dataset for model fine-tuning

Thank you for clarifying. With this in mind I do see the value of dataset and evaluation. I still feel that hierarchical evaluation format / questions and dataset size are limiting, and that there is no need to have a universal VLM for all clinical diagnostic image tasks (that isn't at least fine tuned on a task specific dataset). But I now understand that your benchmark is a first step towards verifying (or falsifying) this feeling of mine.

Your approach of choosing the one "most informative" 2D image out of 3D or 4D volumes is OK at this point, given the limitations of current VLM technology. But please point out that this is a severe shortcoming of all efforts in the community, and will have to be addressed in the future. (I see the other reviewer had similar concerns.)

Please provide the mapping of your training and testing data to the original sources, so that anyone who is fine-tuning on those datasets is not fine-tuning on test data. It would also be valuable if you could flag those models in Table 4 that are likely to have seen any of your public data sources during training (based on the description of their training dataset, if available, or the data use licenses of the datasets themselves).Please add a brief discussion of this matter to the limitations.

**Limitations Weaknesses:**

* I feel that 5,500 images and 1.5 times as many question-answer pairs is a very small number on which to fine-tune an LLM.

* It is not clear what questions were asked at each of the five levels, or how the ground truth was generated. I read somewhere that experts were involved, but who has sufficient expertise in pathology and MRI and US and for a broad rang diagnostic tasks as well as data quality across all scanners and modalities?

* The most important category is 'clinical interpretation'. How was this ground truth generated? Most of the underlying public datasets don't offer much detail. Did experts add this information? Without any context, just by inspecting a 2D image?

*I feel that relying on 2D images is inappropriate for all 3D tasks (MRI, CT and US video). Too much context is missing. Moreover, how was this one slice selected? And by whom? Other question arise and no details are given, e.g.: In the case of an MRI scan, which modality/channel was chosen, or were all channels used? In pathology, which 256 x 256 subset was chosen? Or was the original ~10k*10k image subsampled?

* Many of the questions from levels 1 to 3 can be extracted directly from the metadata of these images. What is the purpose of extracting this information via an LLM? Similarly, FDA-approved, disease-specific lesion detection or classification algorithms are available. What is the purpose of training them via general-purpose LLMs that are much worse? (Don't get me wrong; I can see potential applications, but you should explain them to us.)

* Tell us about the 'key features' extracted via DeepSeek (page 6). Can you also translate the BERT scores into interpretable performance scores for disease/modality/quality prediction? At least for the top-performing algorithms? After all, we are interested in actionable classification, detection and labelling tasks, not measuring linguistic similarities. If your dataset is to be used for comparison in future publications, you may want to make code for your BERT score available?

* What is the 'accuracy' in Figure 4, the BERT score mentioned above?

* I do not see where the "traces of human-like reasoning" are found, and no direct answer to the question of "Do VLMs reason like human doctors in medical image diagnostics" (see suggested by the title) is offered.

**Strengths Contributions:**

* The authors address an important question: How should information in medical image data be structured, and, more importantly, how can algorithms be trained to reason like human experts, i.e. pathologists, radiologists or other medical specialists (e.g., cardiologists using ultrasound)?

* The authors introduce a hierarchical set of categories that can inform general-to-specific reasoning.

* The authors provide a thorough comparison of LLMs.

* The datasets are well presented on Kaggle, Huggingface and GitHub. Routines for using the fine-tuned LLMs are available. (However, individual images are access-restricted, so I cannot check. I am not sure where the QA pairs are available; eventually, some restricted access .json file may contain them.)

---

> ### Author Rebuttal · Authors · 2025-07-31
>
> We thank you for the detailed review and feedback. We value your positive feedback on our benchmark, reasoning framework, and open-source efforts, as well as your suggestions on metrics, human-like reasoning and important insights for improving our work.
>
> We would like to clarify a key point first: DrVD-Bench is intended as a benchmark to evaluate the diagnostic reasoning ability of pretrained VLMs, rather than a dataset for model fine-tuning. All models in our study were evaluated in a zero-shot setting. We hope this clarification helps, and we will address your insightful comments point by point below.
>
> **Response to Limitations Weaknesses 1 (LW1)**
>
> Thank you for raising this point. We would like to clarify that our study are specifically designed for evaluation purposes, not for fine-tuning. All models evaluated in this work in a zero-shot setting without any additional training.
>
> While the dataset size may seem modest compared to pretraining corpora, we believe that it is substantial by benchmarking standards, especially given the complexity of tasks and the involvement of extensive expert validation. Furthermore, compared with recent multimodal medical benchmarks MedXpertQA [1] (4,460 multimodal questions) and II-Bench [2] (1,222 images with up to three QA pairs each), DrVD-Bench offers a larger and more clinically structured evaluation set (5,442 images and 7,789 expert-validated QA pairs), demonstrating that our work is competitive in scale.
>
> [1] Zuo, et al. MedXpertQA: Benchmarking expert-level medical reasoning and understanding. ICML (2025).
>
> [2] Liu, et al.. II-Bench: An Image Implication Understanding Benchmark for Multimodal Large Language Models. NeurIPS (2024).
>
> **Response  to LW2**
>
> Thank you for raising this important concern. DrVD-Bench is structured into five task levels, reflecting the clinical reasoning workflow, with task types detailed in Section 3.2.2 and Figure 2. All questions use a multiple-choice, zero-shot format to ensure consistent evaluation across models.
>
> We would like to clarify that the ground truth for a QA pair is not manually annotated from scratch. Instead, we used scripts to extract them from metadata/annotations in public datasets, followed by expert review. A panel of five medical specialists—three MD candidates from Tsinghua University School of Medicine (including one of our co-authors) and two clinicians from Tsinghua Changgung Hospital and Peking Union Medical College Hospital—systematically verified each QA pair for clinical relevance, visual grounding, and label clarity. As they focused on validation rather than full annotation, this team size ensured both accuracy and scalability.
>
> **Response  to LW3**
>
> Thank you for this insightful question. The clinical interpretation category in DrVD-Bench includes two task types: diagnosis classification and clinical report generation. For diagnosis classification , ground truth labels were obtained by filtering samples from public datasets that provide reliable diagnostic metadata. For report generation, the reference reports were directly sourced from datasets that include corresponding radiology or pathology reports, such as MIMIC-CXR which contain real-world, clinician-authored text aligned with each image.
>
> As shown in our response to LW2, these ground truths were not manually created or inferred by our team, and no expert was asked to assign diagnoses or write reports. Instead, our panel of medical experts was involved only in the final review stage of QA pairs. More explanations about inspecting 2D images can be found in our response to LW4.
>
> **Response  to LW4**
>
> We agree that 3D or temporal context is important in clinical interpretation. However, most current VLMs do not yet support 3D or video inputs, so we rely on 2D representations for consistent evaluation. To mitigate context loss, we adopted a rigorous, expert-guided frame selection process.
>
> For 3D scans, we first filtered out low-resolution studies (<256×256). Most public datasets already include diagnostically representative slices or channels chosen by their original curators; from these, the same expert panel mentioned above conducted a second-stage review, keeping only the frame that best serves the task.
> For pathology images, source datasets already included diagnostic regions of interest (ROIs) on whole-slide images. Instead of down-sampling the full slides, we extracted 256×256 patches centered around the annotated ROIs, preserving key diagnostic features.
>
> This approach reflects common clinical workflows, where decisions are often based on key slices, and aligns with prior VQA benchmarks like VQA-RAD that use single-slice inputs for question construction.
>
> **Response  to LW5**
>
> DrVD-Bench aims to evaluate whether VLMs can reason directly over medical images without relying on structured metadata or fine-tuning. Levels 1 to 3, while covering basic tasks, assess essential visual grounding skills that underpin higher-order reasoning.
> By organizing tasks from simple to complex, we can systematically assess where VLMs succeed or fail in zero-shot settings. This design helps reveal whether a model’s prediction is supported by a coherent reasoning process or achieved via shortcut correlations.
>
> FDA-approved lesion detectors excel at single tasks but remain rigid and modality-specific, limiting real-world scope. In contrast, generalist medical VLMs are attracting increasing attention for their potential to integrate visual understanding and language-based reasoning, enabling more flexible and explainable decision support. Given this growing interest, it is crucial to benchmark these models systematically. DrVD-Bench provides a framework to evaluate their performance across a wide range of imaging tasks, guiding where such models add value and where improvements are needed.
>
> **Response  to LW6**
>
> Thank you for this valuable question. The “key features” refer to clinical keywords extracted from the generated and reference reports using DeepSeek-V3. guiding BERTScore to better capture diagnostic relevance. We use PubMedBERT-based BERTScore for report evaluation and provide task-specific accuracy metrics for classification-style subtasks.
>
> We agree that BERTScore alone may not sufficiently reflect clinical correctness. To address this, we added FactScore as a factual-consistency metric for generated reports: atomic facts are extracted from both the original reference reports and generated reports using DeepSeek-V3, and GPT-4o is used to compare fact-level consistency. We computed recall (coverage of true facts) and specificity (avoidance of hallucinated facts) for each model.
>
> Preliminary results (Table R1) on representative models show low recall (≤ 0.13) and low specificity (≤ 0.09). This confirms that current VLMs often miss key clinical facts and hallucinate unsupported ones, reinforcing your concern that factual errors may not be captured by text similarity metrics. We plan to include full results in the final version.
>
> **Table R1: FactScore Results: Factual Consistency of Report Generation**
>
> | Model               | Recall (std) | Specificity (std) |
> |---------------------|--------------|-------------------|
> | Gemini 2.5 Pro      | 0.13 (0.16)  | 0.07 (0.10)       |
> | Claude 3.7 Sonnet   | 0.06 (0.11)  | 0.03 (0.05)       |
> | Qwen 2.5 VL 7B      | 0.08 (0.13)  | 0.05 (0.09)       |
> | HuatuoGPT           | 0.13 (0.16)  | 0.09 (0.17)       |
>
>
> **Response  to LW7**
>
> The “Accuracy” reported in Figure 4 refers to the average multiple-choice accuracy across all question types in the Visual Evidence Comprehension Task described in Section 5.1. We apologize for the lack of clarity in the figure caption and will make this more explicit in future revisions to avoid confusion.
>
> **Response  to LW8**
>
> Thank you for raising this important point. We define reasoning as a step-wise process from basic visual recognition to high-level diagnostic interpretation. This can be objectively assessed using multiple-choice VQA questions [3]. We assess this through Independent, Joint, and Multi-turn QA settings.
>
> We said “traces of human-like reasoning” as we found some models demonstrates success on early-stage questions, such as modality recognition or organ identification, that require step-wise integration of visual evidence. As shown in Tables 4–5 and Figure 3, models like GPT-4o outperform random baselines on such tasks but struggle with higher-level questions. These trends suggest partial progress, but also highlight that current models remain far from reliable clinical reasoning.
>
> This point is stated in the original Conclusion (lines 277–279). To clarify, we will revise the manuscript to define “human-like reasoning” more explicitly, and expand examples from multi-turn QA to better illustrate current model limitations.
>
> [3] Zellers, et al. From Recognition to Cognition: Visual Commonsense Reasoning. CVPR (2019).
>
> **Response to Dataset Code Comments**
>
> Thank you for raising these concerns. We have re-downloaded the dataset from Kaggle and successfully reproduced all results using our GitHub scripts; other reviewers have reported similar success. For convenience, the dataset is also mirrored on Hugging Face. If you encounter any issues, please share the details—we’re happy to assist in the discussion forum.
> Regarding image traceability: since all images are used solely for evaluation, not training, strict per-image provenance is not required. All source datasets are listed in Table 6, but we’re happy to provide a full mapping file upon request after the review period.
>
> **Response to Additional Feedback**
>
> Thank you for the suggestion. Our benchmark currently focuses on 2D inputs, as most medical VLMs and public QA datasets are designed around representative slices rather than full 3D volumes. We agree that 3D inputs offer richer context and plan to support volume-level evaluation in future iterations. This roadmap will be noted in the revised manuscript.

---

### Note · Authors · 2025-08-16

We sincerely thank all reviewers for their valuable feedback throughout the review and discussion phases. These comments helped us better clarify the positioning of our work and guided us in strengthening key aspects such as data construction, clinical validation, and transparency.

In summary, we have systematically addressed the major points raised and made the following improvements:

1. Clinical validity of data: We clarified that all task labels and diagnostic reports in DrVD-Bench are derived from public datasets and have been systematically reviewed by a panel of medical experts to ensure clinical reliability.

2. Data source transparency: We summarized the training datasets used by medical-specific VLMs and will release a full training–testing mapping file after the review period, improving traceability and helping mitigate potential data leakage.

3. Data bias statement: We acknowledged that current limitations, such as the use of 2D slices instead of full 3D/4D scans and the underrepresentation of minority and resource-limited populations, may impact generalizability. These issues will be further discussed in the revised manuscript’s conclusion.

4. Evaluation enhancements: We enriched the benchmark with several new experiments to improve evaluation depth. Specifically, we introduced FactScore to directly assess factual consistency in generated reports, conducted multi-turn reasoning experiments to investigate error accumulation across steps, and added Patho-R1 to benchmark pathology-specific performance and its limitations in cross-modality generalization.

5. Reproducibility improvements: To facilitate adoption and future extension, we modularized the evaluation toolkit and released it as a PyPI package, accompanied by comprehensive tutorials and third-party validation in fresh environments.

We believe these efforts significantly improve the benchmark’s transparency, usability, and community value. We are grateful for the reviewers’ engagement and hope that this benchmark can serve as a foundation for continued progress in clinical VLM research.

---

### Decision · Program_Chairs · 2025-09-18

**Decision:**

Accept (poster)

**Comment:**

Summary: After the review period, all reviewers (iJcf, 642U, hVkD, and 36gY) award this submission scores between 4 and 6 for an overall average of 4.75 (confidence-unweighted). In a reviewer's (iJcf) words: "I believe [the limitations are] more a reflection of the challenge of constructing the 'perfect' benchmark for medical reasoning, and second Reviewer 36gY's comment that the authors performed a great job and commendable effort putting together a benchmark that may not be 'perfect,' but is still promising and may be of value to the greater community given additional time." Overall, all major points are (at least partially) addressed, and I recommend accept as a poster (or bumped up to spotlight/oral, as seen fit by the SAC and PCs).

Strengths: The submission thoroughly addresses important questions in the field, and providing this benchmark (for the modalities offered) offers the community a strong first-pass test to evaluate understanding/reasoning output based on (clinically taken) images. By structuring the benchmark in a hierarchical fashion (including image modality and quality understanding in and of itself), the outputs become more accessible to the reader, and specific shortcomings of the tested models are revealed. "The organ and leisure erasure tasks are novel to the best of my knowledge." (from reviewer iJcf)

Weaknesses: Some points raised by reviewers (potentially too low number of image/question pairs for full coverage of desirable diagnoses and complexity of clinical applications, reviewer J5rP; remaining questions around some of the ground-truth label data, reviewer J5rP; interpretability of BERT score; some remaining questions around the impact of incorrect inferences within a hierarchical framework, reviewers hVkD and 36gY; the lack of discussion of/comparison with commercially available models due to a number of reasons outside the control of the authors, reviewer hVkD; the utility of certain "tasks", reviewer iJcF) remain not fully addressed, mostly due to the nature of the benchmark and other factors outside of the control of the authors. That being said, the authors did address the bulk of the limitations w.r.t. the value of the presented benchmark for the community. I.e., while these limitations remain largely unaddressed on technical grounds, the authors present a valid argument why those limitations do not take away from the overall value of the work for the community.

Featured reviewer comments:
- J5rP: "The authors introduce a hierarchical set of categories that can inform general-to-specific reasoning."
- J5rP: "Similarly, FDA-approved, disease-specific lesion detection or classification algorithms are available. What is the purpose of training them via general-purpose LLMs that are much worse?"
- hVkD: "DrVD-Bench covers diverse diagnostic stages, from image quality to final diagnosis, allowing fine-grained performance analysis across reasoning levels."
- 36gY: "The analysis perspectives are diverse, and the experiments are thorough. Especially, the part of 'overdiagnosis without understanding' presents key insights."
- 36gY: "The evaluation metrics for reports are too simplistic and lack assessment of hallucinations, which may lead to the frequent use of visual evidence comprehension tasks as benchmarks while report generation evaluation is overlooked."
- iJcF: "The organ and leisure erasure tasks are novel to the best of my knowledge."
- iJcF: "[...] multiple choice QA is generally a poor proxy for clinical reasoning."

Discussion: While the reviewers did not all provide feedback on the rebuttal, scores were broadly increased from the initial values, suggesting that reviewers did take the rebuttals into account.